# Temporal Query Network for Efficient Multivariate Time Series Forecasting

**Shengsheng Lin** [1]   **Haojun Chen** [1]   **Haijie Wu** [1]   **Chunyun Qiu** [1]   **Weiwei Lin** [1 2]

## Abstract

Sufficiently modeling the correlations among variables (aka channels) is crucial for achieving accurate multivariate time series forecasting (MTSF). In this paper, we propose a novel technique called Temporal Query (TQ) to more effectively capture multivariate correlations, thereby improving model performance in MTSF tasks. Technically, the TQ technique employs periodically shifted learnable vectors as queries in the attention mechanism to capture global inter-variable patterns, while the keys and values are derived from the raw input data to encode local, sample-level correlations. Building upon the TQ technique, we develop a simple yet efficient model named Temporal Query Network (TQNet), which employs only a single-layer attention mechanism and a lightweight multi-layer perceptron (MLP). Extensive experiments demonstrate that TQNet learns more robust multivariate correlations, achieving state-of-the-art forecasting accuracy across 12 challenging real-world datasets. Furthermore, TQNet achieves high efficiency comparable to linear-based methods even on high-dimensional datasets, balancing performance and computational cost. The code is available at: https://github.com/ACAT-SCUT/TQNet.

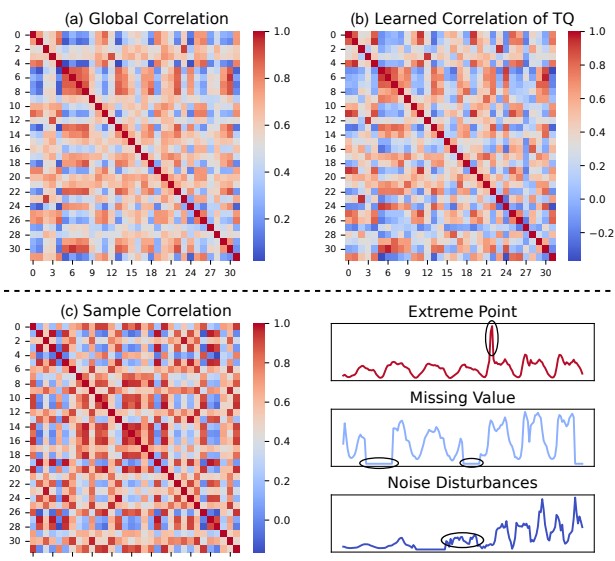

Figure 1: Comparison of inter-variable correlation patterns on the Traffic dataset. (a) Global correlations computed across the entire training set; (b) Correlations learned by the proposed TQ technique; (c) Correlations from individual samples, which may be distorted by non-stationary disturbances such as extreme values, missing data, and noise. The comparison between panels (a) and (b) shows that TQ effectively captures global correlation structures, while panel (c) illustrates the instability of sample-level correlations.

## 1. Introduction

Effectively modeling correlations among variables (aka channels) is crucial for multivariate time series forecasting (MTSF), a task that plays a critical role in various real-world applications, including energy planning, traffic monitoring, medical forecasting, and climate modeling (Qiu et al., 2024; Wang et al., 2024b; Wen et al., 2023; Luo et al., 2024). The accuracy of MTSF relies heavily on capturing precise inter-variable correlations, as these correlations significantly influence the quality of predictions (Wang et al., 2024c).

However, accurately identifying the correlation patterns among multivariate input sequences is inherently challenging, particularly when the sequences are affected by non-stationary disturbances, such as extreme values, missing data, and noisy perturbations (Huang et al., 2023). As illustrated in Figure 1 (a) and (c), these disturbances can create significant discrepancies between the inter-variable correlations observed in individual samples and the global correlations derived from the entire training set. Bridging the gap between local (sample-level) and global (dataset-level) correlation patterns is essential for improving predictive performance (Cai et al., 2024).

To address this issue, we propose the Temporal Query (TQ)

---

[1]School of Computer Science and Engineering, South China University of Technology, Guangzhou 510006, China [2]Pengcheng Laboratory, Shenzhen 518066, China. Correspondence to: Weiwei Lin <linww@scut.edu.cn>.

*Proceedings of the 42nd International Conference on Machine Learning*, Vancouver, Canada. PMLR 267, 2025. Copyright 2025 by the author(s).

technique, which integrates global and local correlations within the attention mechanism. It uses *periodically shifted learnable vectors* as queries to emphasize stable, global dependencies across variables, while the raw input sequences serve as keys and values to capture instance-specific (i.e., local) relationships. The TQ technique offers two key advantages: its learnable nature enables adaptive modeling to capture optimal representations of inter-variable relationships, while its periodic time-shifting mechanism facilitates parameter reuse, resulting in more robust and stable correlation learning. As shown in Figure 1 (a) and (b), the correlations of queries captured by the learned TQ align more closely with the global correlations of the training set, enabling the model to establish stable dependencies during prediction and achieve more accurate forecasts.

Building upon the strong capability of the TQ technique in modeling inter-variable correlations, we propose a simple yet effective multivariate forecaster, termed Temporal Query Network (**TQNet**). TQNet consists of only a single-layer multi-head attention (MHA) mechanism and a shallow multi-layer perceptron (MLP). Extensive experiments demonstrate that TQNet achieves state-of-the-art performance on 12 real-world multivariate datasets by effectively capturing robust inter-variable dependencies. Moreover, its lightweight architecture enables TQNet to maintain high computational efficiency comparable to linear-based methods (e.g., DLinear (Zeng et al., 2023)), even on high-dimensional datasets, while delivering superior forecasting accuracy.

In summary, the contributions of this paper are as follows:

- We propose a novel Temporal Query (TQ) technique, which models global inter-variable correlations within the attention mechanism using periodically shifted learnable vectors as queries.

- Based on the TQ technique, we introduce TQNet, a simple yet powerful model that effectively captures multivariate and temporal dependencies using only a single-layer multi-head attention mechanism and a shallow MLP. This lightweight architecture ensures TQNet achieves efficiency comparable to linear-based models across datasets of varying dimensions.

- Extensive experiments on 12 challenging real-world multivariate datasets demonstrate that TQNet achieves overall state-of-the-art performance by extracting robust inter-variable correlations.

## 2. Related Work

In recent years, deep learning-based approaches have made significant progress in MTSF tasks (Wang et al., 2024b; Qiu et al., 2024; Shao et al., 2024; Lin et al., 2025). These advances reflect a transformation in the strategies for modeling multivariate dependencies, which can be categorized as follows:

**Channel Mixing (CM):** CM-based methods treat each time step as an input unit, meaning that multiple observed values (i.e., variables) at a given time step are mixed and embedded together. This approach is adopted by many early Transformer-based models (Vaswani et al., 2017), such as LogTrans (Li et al., 2019), Informer (Zhou et al., 2021), Autoformer (Wu et al., 2021), FEDformer (Zhou et al., 2022), ETSformer (Woo et al., 2022), and Pyraformer (Liu et al., 2021). However, such methods often make variable relationships indistinguishable within the model (Liu et al., 2024c). As a result, even with their large parameter sizes, these models sometimes fail to outperform simple linear-based baselines (Zeng et al., 2023; Lin et al., 2024a; Toner & Darlow, 2024).

**Channel Independence (CI):** To address the limitations of CM-based methods, PatchTST (Nie et al., 2023) introduced the concept of channel independence (CI). This strategy separates the modeling of each variable and uses a parameter-sharing unified model to make independent forecasts for each channel. Although these methods reduce the capacity for modeling inter-variable dependencies, they enhance model robustness, which is often more critical for time series forecasting tasks (Han et al., 2024b). Consequently, CI-based methods, including SegRNN (Lin et al., 2023), TiDE (Das et al., 2023), N-HiTS (Challu et al., 2023), FITS (Xu et al., 2024), STID (Shao et al., 2022), TimeMixer (Wang et al., 2024a), PDF (Dai et al., 2024), CATS (Kim et al., 2024), SparseTSF (Lin et al., 2024a), and CycleNet (Lin et al., 2024b), have seen widespread adoption in recent years. Nonetheless, their lack of explicit inter-variable relationship modeling has become a bottleneck, limiting their performance in multivariate scenarios.

**Channel Dependence (CD):** To overcome the limitations of CI-based methods, researchers have developed advanced techniques to model inter-variable dependencies. These methods often treat segments of time series or entire channels as input units, or employ advanced mechanisms to better understand multivariate relationships. Such approaches include various attention-based methods, such as TimeXer (Wang et al., 2024c), iTransformer (Liu et al., 2024c), Crossformer (Zhang & Yan, 2023), CARD (Xue et al., 2024), SAMformer (Ilbert et al., 2024), Leddam (Yu et al., 2024), UniTST (Liu et al., 2024a), SSCNN (Deng et al., 2024), TimeBridge (Liu et al., 2024b), and DUET (Qiu et al., 2025). Additionally, other paradigms have also been proposed to tackle this challenge. These include graph neural networks (e.g., CrossGNN (Huang et al., 2023), FourierGNN (Yi et al., 2023)), convolutional neural networks (e.g., MICN (Wang et al., 2023), TimesNet (Wu et al., 2023), Mod-

ernTCN (Luo & Wang, 2024)), and MLP-based solutions (e.g., LightTS (Zhang et al., 2022), MTS-Mixers (Li et al., 2023b), TSMixer (Chen et al., 2023; Ekambaram et al., 2023), HDMixer (Huang et al., 2024), SOFTS (Han et al., 2024a)). These methodologies demonstrate diverse and effective ways to capture and model the intricate dependencies among multivariate time series.

In this paper, we focus on attention-based approaches within the scope of CD-based methods and aim to design a minimalist yet efficient network for MTSF. Inspired by the periodicity modeling in CycleNet (Lin et al., 2024b), which employs learnable parameters to capture period patterns, we propose the Temporal Query (TQ) technique. The TQ technique introduces periodically shifted learnable vectors as query inputs to the attention mechanism, enabling the model to incorporate inherent periodic structures in the data while more effectively capturing robust inter-variable relationships.

## 3. Methodology

The goal of multivariate time series forecasting (MTSF) is to predict future sequences $Y_t \in \mathbb{R}^{C \times H}$ based on observed historical sequences $X_t \in \mathbb{R}^{C \times L}$ at time step $t$, where $C$ represents the number of variables (or channels), $H$ is the forecasting horizon, and $L$ is the length of the look-back window. To address the MTSF task, the workflow of the proposed Temporal Query Network (TQNet) is illustrated in Figure 2, with the corresponding pseudocode provided in Appendix A.1.

### 3.1. Overview of TQNet

TQNet consists of a series of simple yet essential components. Specifically, given an input sequence $X_t \in \mathbb{R}^{C \times L}$, the temporal query-enhanced multi-head attention (TQ-MHA) layer is first applied to capture multivariate correlations. This is followed by a shallow multi-layer perceptron (MLP), which models temporal dependencies. Both the TQ-MHA and MLP are enhanced with residual connections (He et al., 2016) to improve learning stability. Finally, a linear layer with Dropout (Srivastava et al., 2014) is employed to project the learned hidden representations onto the target outputs $\bar{Y}_t \in \mathbb{R}^{C \times H}$. Below, we detail the key components of TQNet.

### 3.2. Components of TQNet

**Temporal Query Technique** The Temporal Query (TQ) technique uses periodically shifted learnable vectors as queries in the attention mechanism to capture global inter-variable correlations (see Figure 3). Mathematically, given the periodic length $W$ of the dataset, we initialize $\theta_{\text{TQ}} \in \mathbb{R}^{C \times W}$, which serves as a set of learnable parameters. These

parameters are initialized to zeros and dynamically updated during training to adaptively capture the underlying correlations among variables.

Herein, the hyperparameter $W$ determines the length of the learnable vectors and the interval for periodic shifts. Its value should align with the stable periodicity of the dataset, which can be identified either through prior knowledge about the data (Lin et al., 2024b) or via computational methods, such as the autocorrelation function (ACF) (Madsen, 2007). In Section 4.3, we will provide a detailed discussion on the criteria for selecting $W$ and its influence on the performance of the TQ technique.

**TQ-Enhanced Multi-Head Attention (TQ-MHA)** To effectively capture multivariate dependencies, TQNet employs a single-layer multi-head attention (MHA) mechanism enhanced by the proposed TQ technique. Conventional self-attention mechanisms, such as those in iTransformer (Liu et al., 2024c), derive the queries, keys, and values ($Q$, $K$, $V$) directly from the input sample sequences. However, as discussed earlier, non-stationary disturbances in real-world data, such as noise or extreme values, can impede the accurate modeling of correlations within individual samples. To address this, we propose a TQ-enhanced MHA mechanism, where the queries are generated from the TQ vectors to capture global correlations:

$$Q = \theta_{\text{TQ}}^{t,L} \in \mathbb{R}^{C \times L}. \tag{1}$$

Here, $\theta_{\text{TQ}}^{t,L}$ is a segment of the learnable parameters $\theta_{\text{TQ}} \in \mathbb{R}^{C \times W}$, extracted periodically for each input sample. Specifically, for a given time step $t$, the starting index is computed as $t \bmod W$, and a segment of length $L$ is cyclically selected from $\theta_{\text{TQ}}$ (as shown in Figure 3). This implies that for samples spaced $W$ time steps apart, the extracted TQ vectors will remain identical, which can be expressed as:

$$\theta_{\text{TQ}}^{t,L} = \theta_{\text{TQ}}^{(t+i \cdot W),L}, \quad i \in \mathbb{N}. \tag{2}$$

The periodic shifting mechanism ensures that $\theta_{\text{TQ}}^{t,L}$ cycles periodically over time, aligning with the prior knowledge of periodic variations in real-world data (Lin et al., 2024b). This mechanism also facilitates efficient parameter reuse, enabling the modeling of more robust sequence correlations by mitigating (i.e., averaging) the impact of localized noisy perturbations. By integrating TQ into the MHA mechanism, the attention process for each head $h$ becomes:

$$\text{Head}_h = \text{Softmax}\left(\frac{Q_h K_h^\top}{\sqrt{L}}\right) V_h, \tag{3}$$

where $Q_h = \theta_{\text{TQ}}^{t,L} W_h^Q$, $K_h = X_t W_h^K$, and $V_h = X_t W_h^V$. After computing the attention outputs for all heads, the

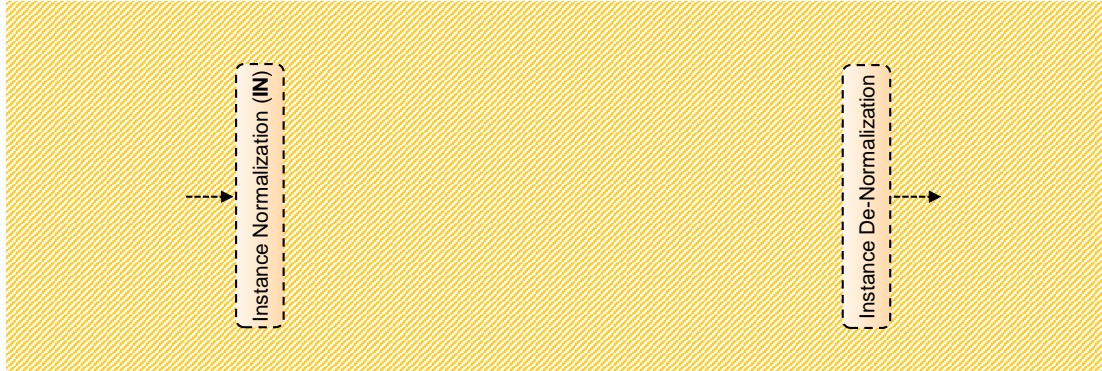

Figure 2: The architecture of the proposed TQNet model. It comprises the lightweight and efficient TQ-MHA and MLP modules, with an optional IN module to mitigate distributional drift.

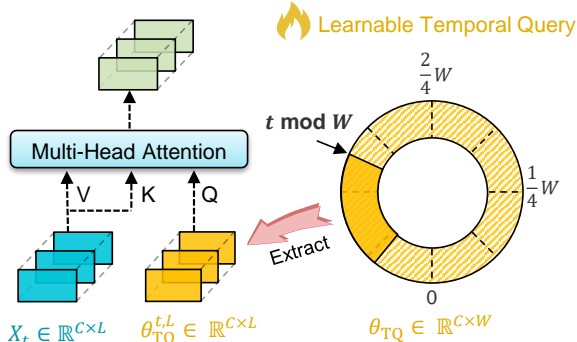

Figure 3: Temporal Query-enhanced Multi-Head Attention (TQ-MHA). The proposed TQ technique employs periodically shifted learnable parameters as queries to adaptively model inter-variable correlations within individual samples.

MHA mechanism concatenates the outputs:

$$\text{MHA}(Q, K, V) = \text{Concat}(\text{head}_1, \ldots, \text{head}_H)W^O, \quad (4)$$

where $W^O$ is the output projection matrix of MHA module.

By incorporating $\theta_{\text{TQ}}^{t,L}$ as query vectors, the TQ-enhanced MHA mechanism fuses global priors with local input features, enabling more robust modeling of inter-variable correlations. Specifically, the periodic design of $\theta_{\text{TQ}}^{t,L}$ captures stable global patterns (as shown in Figure 1(b)), while keys and values from $X_t$ retain sample-specific information (as shown in Figure 1(c)). This design improves resilience to noise and missing data by balancing global consistency and local adaptability.

**Multi-Layer Perceptron (MLP)**  Following the TQ-enhanced attention mechanism, a lightweight MLP is employed to model temporal dependencies within the sequence. Previous studies have demonstrated the robustness of MLPs in extracting temporal features (Li et al., 2023a). The MLP

in TQNet consists of two fully connected layers interspersed with GeLU activations (Hendrycks & Gimpel, 2016), defined as:

$$h_{\text{mlp}} = \text{Linear}(\text{GeLU}(\text{Linear}(h_{\text{attn}}))), \quad (5)$$

where $h_{\text{attn}}$ represents the output of the TQ-MHA module.

**Output Projection**  The output of the MLP is fed into a linear layer, optionally with Dropout (Srivastava et al., 2014), to project the learned representations onto the target forecasting horizon:

$$\bar{Y}_t = \text{Linear}(\text{Dropout}(h_{\text{mlp}})). \quad (6)$$

**Instance Normalization (IN)**  Distributional shifts are common in real-world data and can significantly impact the generalization performance of models. To address this challenge, recent research has explored various approaches, such as RevIN (Kim et al., 2021) and FAN (Ye et al., 2024). In this paper, we adopt a simple yet effective IN method that used in iTransformer (Liu et al., 2024c) and CycleNet (Lin et al., 2024b), which involves removing the mean and variance of the input data before and after the model's operation:

$$X_t = \frac{X_t - \mu}{\sqrt{\sigma^2 + \epsilon}}, \quad (7)$$

$$\bar{Y}_t = \bar{Y}_t \times \sqrt{\sigma^2 + \epsilon} + \mu, \quad (8)$$

where $\mu$ and $\sigma^2$ denote the mean and variance of the input data, and $\epsilon$ is a small constant added for numerical stability.

## 4. Experiments

### 4.1. Setup

The experiments in this section are implemented using Py-Torch (Paszke et al., 2019), with the models optimized using L2 loss and evaluated based on both Mean Squared Error

(MSE) and Mean Absolute Error (MAE). Further details about the experimental setup can be found in Appendix A.2. The datasets and baseline models used for evaluation are outlined as follows.

Table 1: Detailed information about the datasets. The hyper-parameter $W$ of TQ technique is configured to align with the stable cycle length of each dataset, following the guidelines in CycleNet (Lin et al., 2024b).

| Dataset | Channels | Timesteps | Interval | HParam. $W$ | Domain |
|---|---|---|---|---|---|
| ETTh1 | 7 | 14,400 | 1 hour | 24 | Electricity |
| ETTh2 | 7 | 14,400 | 1 hour | 24 | Electricity |
| ETTm1 | 7 | 57,600 | 15 mins | 96 | Electricity |
| ETTm2 | 7 | 57,600 | 15 mins | 96 | Electricity |
| Electricity | 321 | 26,304 | 1 hour | 168 | Electricity |
| Solar | 137 | 52,560 | 10 mins | 144 | Energy |
| Traffic | 862 | 17,544 | 1 hour | 168 | Transportation |
| Weather | 21 | 52,696 | 10 mins | 144 | Weather |
| PEMS03 | 358 | 26,208 | 5 mins | 288 | Transportation |
| PEMS04 | 307 | 16,992 | 5 mins | 288 | Transportation |
| PEMS07 | 883 | 28,224 | 5 mins | 288 | Transportation |
| PEMS08 | 170 | 17,856 | 5 mins | 288 | Transportation |

**Datasets** We evaluate the proposed method on 12 widely used real-world datasets, including the ETT series (Zhou et al., 2021), PEMS series (Liu et al., 2022a), Electricity, Solar, Traffic, and Weather datasets (Wu et al., 2021). These datasets vary in scale, dimensionality (i.e., the number of variables), frequency, and domain. Detailed information about the datasets is provided in Table 1.

**Baselines** To evaluate the performance of the proposed TQNet, we compare it against several representative models from recent years, including TimeXer (Wang et al., 2024c), CycleNet (Lin et al., 2024b), iTransformer (Liu et al., 2024c), MSGNet (Cai et al., 2024), TimesNet (Wu et al., 2023), PatchTST (Nie et al., 2023), Crossformer (Zhang & Yan, 2023), DLinear (Zeng et al., 2023), SCINet (Liu et al., 2022a). Following the setting of iTransformer, TQNet defaults to a look-back length of 96.

### 4.2. Main Results

Table 2 compares the forecasting errors of TQNet with baseline models across 12 real-world datasets. Lower MSE and MAE values indicate higher forecasting accuracy. TQNet consistently achieves Top 2 performance in 22 out of 24 forecasting error metrics, demonstrating overall state-of-the-art accuracy. Notably, TQNet excels in high-dimensional multivariate datasets, such as Electricity and PEMS (over 100 variables). This highlights TQNet's advantages in handling complex multivariate data.

In addition, iTransformer and TimeXer, which are also advanced attention-based methods, exhibit strong performance in complex multivariate scenarios. However, as mentioned

earlier, these methods rely entirely on self-attention to extract dependencies, making them more vulnerable to noisy disturbances in input samples. This susceptibility limits their potential for further improvements in predictive performance. Conversely, methods like PatchTST and DLinear, which are CI-based approaches lacking explicit mechanisms for modeling inter-variable dependencies, exhibit suboptimal performance on high-dimensional multivariate datasets.

Overall, despite the simplicity of TQNet's architecture, it achieves superior performance. This significant improvement is primarily attributed to the TQ technique, which effectively enhances the model's ability to capture complex multivariate correlations. In the following sections, we delve deeper into the contributions of the TQ technique in modeling multivariate dependencies.

### 4.3. Ablation Studies and Analysis

**Ablation Study on the Design of TQ-MHA** The goal of the proposed TQ technique is to better integrate both global and local correlations within the attention mechanism. Specifically, the queries $Q$ are generated from periodically shifted learnable vectors $\theta_{\text{TQ}}$, while the keys $K$ are derived from the raw input sequences. To examine the effects of global versus local correlations, we compare the following three settings:

1. Both $Q$ and $K$ are generated from raw data, capturing only per-sample correlations. This is the conventional approach used in models such as iTransformer and TimeXer.

2. $Q$ is generated from the learnable TQ vectors, while $K$ is derived from the raw data. This enables the attention score computation (i.e., $\frac{QK^{\top}}{\sqrt{L}}$) to incorporate both global and sample-specific correlations. This corresponds to the design used in our TQNet.

3. Both $Q$ and $K$ are generated from the learnable TQ vectors, thereby focusing the attention mechanism solely on global correlations, ignoring local variations.

Table 3 presents the average performance across four forecasting horizons on large-scale multivariate datasets (those with more than 100 variables). As shown, the TQNet strategy (i.e., both global and per-sample correlations are considered) yields the best performance. This is followed by the setting that considers only per-sample correlations, and then the purely global setting. These results validate that a proper integration of both global and local correlations is indeed beneficial for multivariate forecasting.

**Ablation Study on TQNet Components** To further investigate the individual contributions of TQNet's components, we conduct ablation experiments on high-dimensional

Table 2: Comparison of multivariate time series forecasting results across 12 real-world datasets. The reported results are averaged over all prediction horizons $H \in \{96, 192, 336, 720\}$, with detailed results available in Table 5. The look-back length $L$ is uniformly fixed at 96. The best results are highlighted in **bold**, the second best are underlined, and the *Count* row counts the number of times each model ranks in the top 2.

| Model | TQNet (Ours) | | TimeXer (2024c) | | CycleNet (2024b) | | iTransformer (2024c) | | MSGNet (2024) | | TimesNet (2023) | | PatchTST (2023) | | Crossformer (2023) | | DLinear (2023) | | SCINet (2022a) | |
|---|---|---|---|---|---|---|---|---|---|---|---|---|---|---|---|---|---|---|---|---|
| Metric | MSE | MAE | MSE | MAE | MSE | MAE | MSE | MAE | MSE | MAE | MSE | MAE | MSE | MAE | MSE | MAE | MSE | MAE | MSE | MAE |
| ETTh1 | 0.441 | **0.434** | **0.437** | 0.437 | 0.457 | 0.441 | 0.454 | 0.448 | 0.453 | 0.453 | 0.458 | 0.450 | 0.469 | 0.455 | 0.529 | 0.522 | 0.456 | 0.452 | 0.747 | 0.647 |
| ETTh2 | 0.378 | 0.402 | **0.368** | **0.396** | 0.388 | 0.409 | 0.383 | 0.407 | 0.413 | 0.427 | 0.414 | 0.427 | 0.387 | 0.407 | 0.942 | 0.684 | 0.559 | 0.515 | 0.954 | 0.723 |
| ETTm1 | **0.377** | **0.393** | 0.382 | 0.397 | 0.379 | 0.396 | 0.407 | 0.410 | 0.400 | 0.412 | 0.400 | 0.406 | 0.387 | 0.400 | 0.513 | 0.495 | 0.403 | 0.407 | 0.486 | 0.481 |
| ETTm2 | 0.277 | 0.323 | 0.274 | 0.322 | **0.266** | **0.314** | 0.288 | 0.332 | 0.289 | 0.330 | 0.291 | 0.333 | 0.281 | 0.326 | 0.757 | 0.611 | 0.350 | 0.401 | 0.571 | 0.537 |
| Electricity | **0.164** | **0.259** | 0.171 | 0.270 | 0.168 | 0.259 | 0.178 | 0.270 | 0.194 | 0.301 | 0.193 | 0.295 | 0.205 | 0.290 | 0.244 | 0.334 | 0.212 | 0.300 | 0.571 | 0.537 |
| Solar | **0.198** | **0.256** | 0.237 | 0.302 | 0.210 | 0.261 | 0.233 | 0.262 | 0.263 | 0.292 | 0.301 | 0.319 | 0.270 | 0.307 | 0.641 | 0.639 | 0.330 | 0.401 | 0.282 | 0.375 |
| Traffic | 0.445 | **0.276** | 0.466 | 0.287 | 0.472 | 0.301 | **0.428** | 0.282 | 0.660 | 0.382 | 0.620 | 0.336 | 0.481 | 0.300 | 0.550 | 0.304 | 0.625 | 0.383 | 0.804 | 0.509 |
| Weather | 0.242 | **0.269** | **0.241** | 0.271 | 0.243 | 0.271 | 0.258 | 0.278 | 0.249 | 0.278 | 0.259 | 0.287 | 0.259 | 0.273 | 0.259 | 0.315 | 0.265 | 0.317 | 0.292 | 0.363 |
| PEMS03 | **0.097** | **0.203** | 0.112 | 0.214 | 0.118 | 0.226 | 0.113 | 0.222 | 0.150 | 0.251 | 0.147 | 0.248 | 0.180 | 0.291 | 0.169 | 0.282 | 0.278 | 0.375 | 0.114 | 0.224 |
| PEMS04 | **0.091** | **0.197** | 0.105 | 0.209 | 0.119 | 0.232 | 0.111 | 0.221 | 0.122 | 0.239 | 0.129 | 0.241 | 0.195 | 0.307 | 0.209 | 0.314 | 0.295 | 0.388 | 0.093 | 0.202 |
| PEMS07 | **0.075** | **0.171** | 0.085 | 0.182 | 0.113 | 0.214 | 0.101 | 0.204 | 0.122 | 0.227 | 0.125 | 0.226 | 0.211 | 0.303 | 0.235 | 0.315 | 0.329 | 0.396 | 0.119 | 0.217 |
| PEMS08 | **0.142** | 0.229 | 0.175 | 0.250 | 0.150 | 0.246 | 0.150 | **0.226** | 0.205 | 0.285 | 0.193 | 0.271 | 0.280 | 0.321 | 0.268 | 0.307 | 0.379 | 0.416 | 0.159 | 0.244 |
| Count | **22** | | 11 | | 9 | | 4 | | 0 | | 0 | | 0 | | 0 | | 0 | | 2 | |

Table 3: Performance comparison of different Query-Key configurations in TQ-MHA. The setting ($Q = TQ, K = $ Raw), where $Q$ is generated from global learnable TQ vectors and $K$ is derived from local raw data, corresponds to the default configuration in TQNet.

| Setup | ($Q$=Raw,$K$=Raw) | | ($Q$=TQ,$K$=Raw) | | ($Q$=TQ,$K$=TQ) | |
|---|---|---|---|---|---|---|
| Metric | MSE | MAE | MSE | MAE | MSE | MAE |
| Electricity | 0.175 | 0.267 | **0.164** | **0.259** | 0.179 | 0.269 |
| Solar | 0.208 | 0.257 | **0.198** | **0.256** | 0.213 | 0.268 |
| Traffic | **0.426** | 0.279 | 0.445 | **0.276** | 0.429 | 0.281 |
| PEMS03 | 0.114 | 0.222 | **0.097** | **0.203** | 0.111 | 0.221 |
| PEMS04 | 0.112 | 0.222 | **0.091** | **0.197** | 0.113 | 0.222 |
| PEMS07 | 0.094 | 0.195 | **0.075** | **0.171** | 0.092 | 0.195 |
| PEMS08 | 0.170 | 0.252 | **0.142** | **0.229** | 0.174 | 0.257 |

datasets. Specifically, the core of TQNet consists of an attention mechanism (i.e., MHA) enhanced by the TQ technique for modeling multivariate correlations and a MLP module for capturing temporal dependencies.

Herein, when the TQ technique is removed, TQNet reduces to a standard self-attention module combined with an MLP module, where channel correlations are more susceptible to noisy disturbances. When the attention mechanism is removed, TQNet becomes a variant of an MLP augmented with channel identifiers (Shao et al., 2022), which explicitly distinguish different channels but lack the ability to directly model inter-channel correlations. When both the TQ module and the attention mechanism are removed, TQNet further simplifies into a pure MLP model.

As shown in the results on the left side of Table 4, the TQ technique plays a pivotal role in TQNet's performance, as removing the TQ module leads to the most significant performance degradation. Additionally, the attention mechanism is also critical for modeling inter-variable correlations, as it

enables effective channel-wise information interaction. Finally, the pure MLP model performs the worst due to its lack of any mechanism for channel correlations modeling. As a result, these findings underscore the effectiveness of the TQ technique and demonstrate that its integration with the attention mechanism significantly enhances the performance of multivariate forecasting tasks.

**Integration Study** Beyond validating the critical role of the TQ technique within TQNet, we further explore its portability and effectiveness in improving the multivariate forecasting capabilities of existing models. Specifically, we integrate the TQ technique into several mainstream time series forecasting models to examine whether it enhances their predictive performance.

Notably, i) For iTransformer, a Transformer-based method that uses self-attention to model multivariate correlations, we directly replace the original query inputs with the proposed TQ vectors. ii) For PatchTST and DLinear, which are CI-based methods without dedicated attention mechanisms for modeling channel correlations, we embed a complete attention mechanism (i.e., MHA) incorporating the TQ technique to jointly model channel dependencies.

The results on the right side of Table 4 demonstrate that incorporating the TQ technique (either by enhancing existing attention mechanisms or by embedding a complete attention module with TQ) consistently improves the original models' performance. This indicates the versatility and effectiveness of the TQ technique in elevating the predictive capabilities of diverse forecasting methods, showcasing its portability across different architectures.

**Representation Learning of the TQ Technique** To further explore the underlying factors contributing to the success of TQNet, we examine the intrinsic representations

Table 4: Ablation and integration studies of the TQ technique. Left: Ablation study revealing the effectiveness of the TQ technique within TQNet. Right: Integration study exploring the portability and adaptability of the TQ technique.

| Model | | TQNet (MLP + TQ & MHA) | | | | | | | | iTransformer (2024c) | | | | PatchTST (2023) | | | | DLinear (2023) | | | |
|---|---|---|---|---|---|---|---|---|---|---|---|---|---|---|---|---|---|---|---|---|---|---|
| Setup | | Original | | MHA → None | | TQ → None | | Pure MLP | | Original | | + TQ | | Original | | + TQ & MHA | | Original | | + TQ & MHA | |
| Metric | | MSE | MAE | MSE | MAE | MSE | MAE | MSE | MAE | MSE | MAE | MSE | MAE | MSE | MAE | MSE | MAE | MSE | MAE | MSE | MAE |
| Electricity | 96 | **0.134** | **0.229** | 0.138 | 0.233 | 0.149 | 0.241 | 0.165 | 0.252 | 0.150 | 0.241 | **0.133** | **0.230** | 0.164 | 0.254 | **0.136** | **0.235** | 0.195 | 0.277 | **0.154** | **0.252** |
| | 192 | **0.154** | **0.247** | 0.155 | 0.248 | 0.162 | 0.254 | 0.173 | 0.260 | 0.164 | 0.255 | **0.152** | **0.247** | 0.174 | 0.265 | **0.157** | **0.254** | 0.194 | 0.279 | **0.168** | **0.264** |
| | 336 | **0.169** | **0.264** | 0.172 | 0.267 | 0.178 | 0.271 | 0.190 | 0.277 | 0.178 | 0.272 | **0.166** | **0.264** | 0.193 | 0.285 | **0.177** | **0.277** | 0.207 | 0.295 | **0.181** | **0.279** |
| | 720 | **0.201** | **0.294** | 0.210 | 0.300 | 0.210 | 0.300 | 0.230 | 0.312 | 0.210 | 0.300 | **0.202** | **0.298** | 0.232 | 0.320 | **0.215** | **0.313** | 0.244 | 0.330 | **0.224** | **0.319** |
| | Avg | **0.164** | **0.259** | 0.169 | 0.262 | 0.175 | 0.267 | 0.190 | 0.276 | 0.175 | 0.267 | **0.163** | **0.260** | 0.191 | 0.281 | **0.171** | **0.270** | 0.210 | 0.295 | **0.182** | **0.279** |
| PEMS03 | 12 | **0.060** | **0.161** | 0.061 | 0.164 | 0.065 | 0.168 | 0.071 | 0.177 | 0.106 | 0.219 | **0.067** | **0.166** | 0.072 | 0.179 | **0.068** | **0.168** | 0.105 | 0.221 | **0.100** | **0.185** |
| | 24 | **0.077** | **0.182** | 0.078 | 0.186 | 0.086 | 0.195 | 0.103 | 0.212 | 0.090 | 0.199 | **0.085** | **0.185** | 0.102 | 0.213 | **0.084** | **0.191** | 0.183 | 0.299 | **0.172** | **0.227** |
| | 48 | **0.104** | **0.215** | 0.110 | 0.218 | 0.126 | 0.239 | 0.158 | 0.265 | 0.199 | 0.304 | **0.108** | **0.210** | 0.155 | 0.263 | **0.115** | **0.223** | 0.315 | 0.407 | 0.325 | **0.300** |
| | 96 | **0.148** | **0.253** | 0.157 | 0.255 | 0.174 | 0.284 | 0.208 | 0.309 | 0.242 | 0.348 | **0.156** | **0.248** | 0.204 | 0.305 | **0.155** | **0.263** | 0.455 | 0.508 | 0.532 | **0.382** |
| | Avg | **0.097** | **0.203** | 0.101 | 0.206 | 0.113 | 0.221 | 0.135 | 0.241 | 0.159 | 0.268 | **0.104** | **0.202** | 0.133 | 0.240 | **0.106** | **0.211** | 0.265 | 0.359 | 0.282 | **0.273** |
| PEMS04 | 12 | **0.067** | **0.166** | 0.072 | 0.176 | 0.076 | 0.180 | 0.086 | 0.193 | 0.081 | 0.189 | **0.067** | **0.167** | 0.087 | 0.195 | **0.069** | **0.172** | 0.114 | 0.228 | **0.074** | **0.177** |
| | 24 | **0.077** | **0.181** | 0.086 | 0.194 | 0.094 | 0.204 | 0.119 | 0.229 | 0.097 | 0.207 | **0.079** | **0.182** | 0.119 | 0.231 | **0.082** | **0.189** | 0.187 | 0.298 | **0.093** | **0.202** |
| | 48 | **0.097** | **0.206** | 0.111 | 0.223 | 0.124 | 0.238 | 0.174 | 0.283 | 0.128 | 0.241 | **0.092** | **0.199** | 0.172 | 0.279 | **0.101** | **0.212** | 0.319 | 0.402 | **0.138** | **0.248** |
| | 96 | **0.123** | **0.233** | 0.138 | 0.246 | 0.156 | 0.268 | 0.224 | 0.328 | 0.176 | 0.284 | **0.128** | **0.233** | 0.221 | 0.323 | **0.131** | **0.247** | 0.424 | 0.481 | **0.211** | **0.313** |
| | Avg | **0.091** | **0.197** | 0.102 | 0.210 | 0.113 | 0.223 | 0.151 | 0.258 | 0.120 | 0.230 | **0.091** | **0.195** | 0.150 | 0.257 | **0.096** | **0.205** | 0.261 | 0.352 | **0.129** | **0.235** |

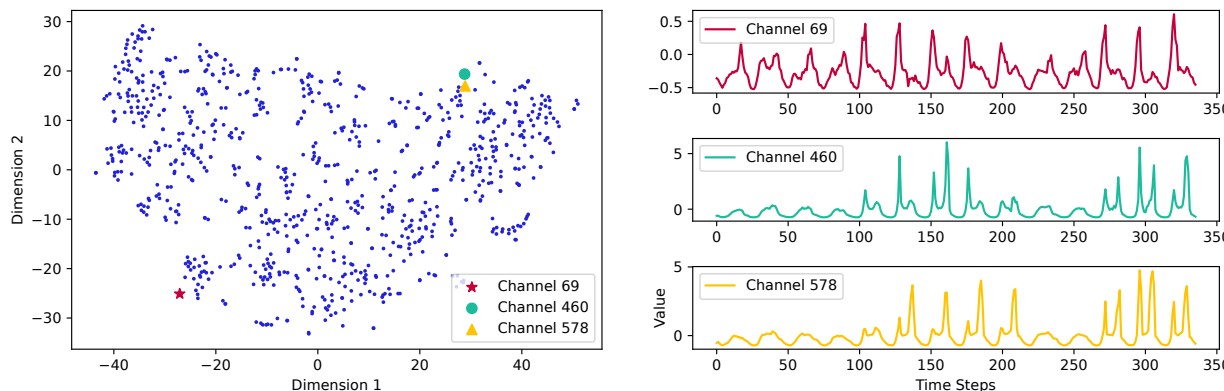

Figure 4: Visualization of the learned TQ representations using t-SNE (Van der Maaten & Hinton, 2008) and their corresponding raw channel sequences. The data is sourced from the Traffic dataset.

learned by the TQ technique. Specifically, Figure 4 presents a visualization of the t-SNE (Van der Maaten & Hinton, 2008) compressed representations of the TQ vectors after TQNet has been trained to convergence.

Remarkably, channels that are closer in the t-SNE representation tend to exhibit similar sequence patterns in their raw time series, whereas channels that are farther apart demonstrate distinct sequence patterns. This suggests that the TQ technique effectively captures intrinsic correlations among different channels. This ability allows the model to leverage information from correlated channels to stabilize predictions for a given channel, particularly in the presence of noisy disturbances. As a result, this characteristic not only improves forecasting accuracy but also enhances the interpretability of the model.

**Dependency Study** In previous experiments, we demonstrated the effectiveness of the TQ technique in improving

forecasting accuracy and learning meaningful semantic representations. In this section, we further investigate whether TQ can capture more robust multivariate dependencies that contribute to better predictive performance.

To this end, we conduct the multivariate-to-univariate forecasting task, where the objective is to predict a single target channel using a varying number of input variables, ranging from no covariate information to utilizing all available variables. As shown in Figure 5, incorporating a moderate amount of covariate information notably enhances the forecasting accuracy of TQNet on the target channel. Overall, using more covariates generally leads to better performance. These results provide direct evidence that TQNet effectively leverages multivariate dependencies to improve prediction quality.

**Impact of the Hyperparameter $W$** In the TQ technique, the length of the learnable TQ vectors, denoted as the hyper-

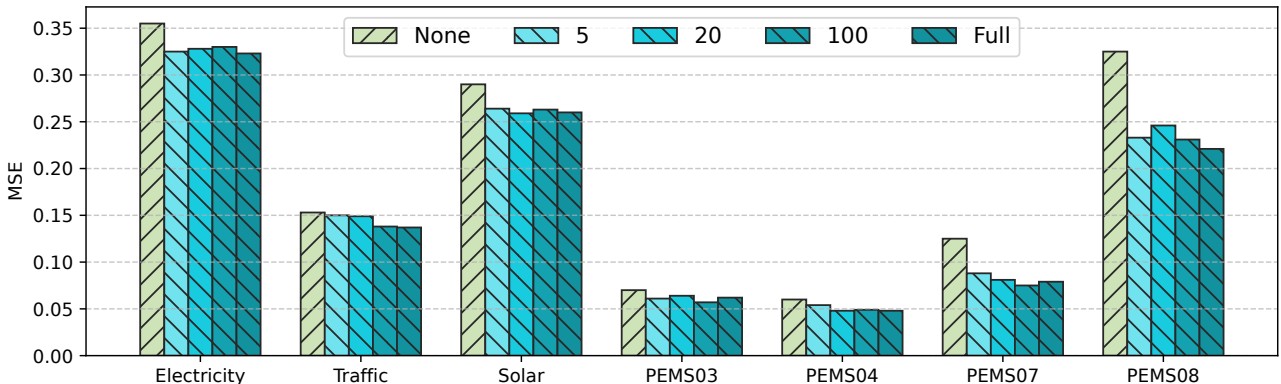

Figure 5: Performance of TQNet with varying amounts of covariate information. The forecasting target is the last channel of the dataset, and the results are averaged across forecasting horizons $H \in \{96, 192, 336, 720\}$. "None" denotes that no covariates are used, "Full" indicates the use of all available covariates, and the numbers represent the use of a partial subset of covariates.

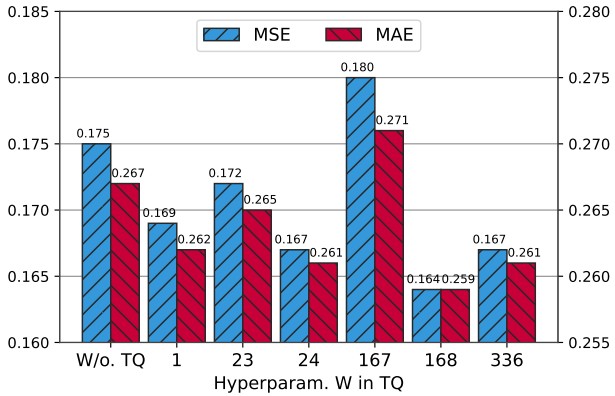

Figure 6: Performance of TQNet on the Electricity dataset with varying hyperparameter $W$. Results are averaged across forecasting horizons $H \in \{96, 192, 336, 720\}$.

parameter $W$, is a critical factor as it determines the periodic shifting interval of the TQ vectors. In principle, this length $W$ should align with the maximum periodic length of the dataset (Lin et al., 2024b), such as the length of a day for daily datasets or the length of a week for weekly datasets, to maximize its effectiveness. To investigate the impact of the hyperparameter $W$, we conduct experiments on the Electricity dataset, which exhibits both daily and weekly periodicity.

As shown in Figure 6, TQNet achieves its best performance when $W$ matches the weekly periodicity of the dataset ($W = 168$[1]). When $W$ matches only the daily periodicity ($W = 24$), TQNet delivers suboptimal perfor-

mance as the model captures only partial periodic features. In addition, when $W$ is set to the length of two weeks ($W = 2 \times 168 = 336$), TQNet still achieves competitive performance. This is primarily because setting $W$ to an integer multiple of the true period merely reduces the number of training samples allocated to each TQ parameter proportionally, without significantly impairing the effectiveness of the TQ mechanism itself.

In contrast, when $W$ does not align with any true periodic length (e.g., $W = 23$ or $W = 167$), the TQ technique introduces semantic inconsistencies, which lead to performance degradation. Additionally, a special case arises when $W = 1$. Although it does not correspond to any periodic cycle, it functions as a weakened form of channel identifier, allowing the model to distinguish between different channels (Shao et al., 2022). This enables TQNet to still achieve performance improvements over the absence of the TQ technique, highlighting its robustness and inherent ability to enhance forecasting accuracy. Finally, we further provide a more detailed discussion in Appendix A.3 on how to determine a proper value of the hyperparameter $W$ and analyze the challenges posed by multi-periodic patterns.

**Efficiency Analysis** Benefiting from the powerful TQ technique and the lightweight architectural design of TQNet, the model achieves exceptional multivariate forecasting performance while maintaining high computational efficiency. As shown in Figure 7 Left, TQNet delivers state-of-the-art forecasting accuracy with significantly smaller parameter sizes and shorter training times compared to other models. This highlights TQNet's ability to balance forecasting performance with computational cost effectively.

Moreover, while TQNet demonstrates impressive capabilities in modeling multivariate correlations, its attention mech-

---

[1]The Electricity dataset is sampled hourly, so the weekly length corresponds to $24 \times 7 = 168$

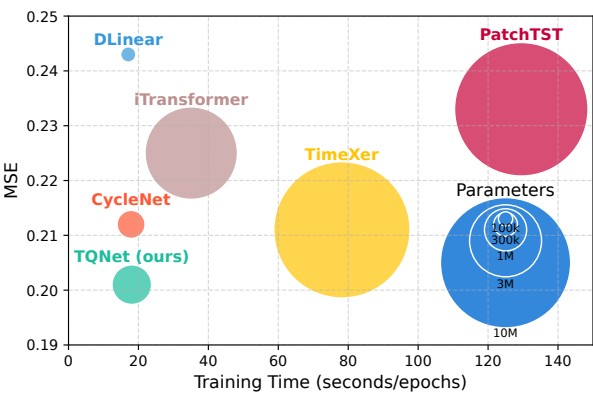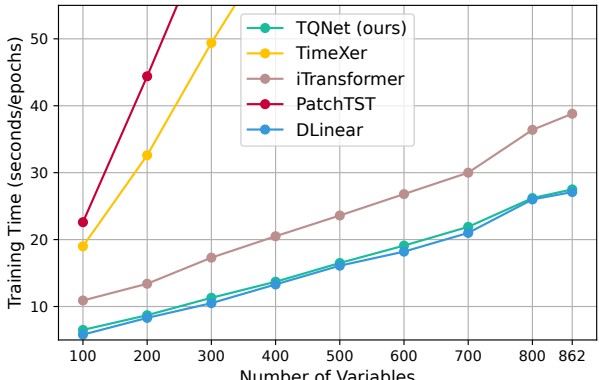

Figure 7: Computational efficiency of TQNet. Left: Comparison of prediction accuracy, parameter size, and training time on the Electricity dataset (forecast horizon $H = 720$). Right: Comparison of training time with varying channel numbers on the Traffic dataset, which includes up to 862 channels in the complete setting.

anism has a quadratic complexity, which may raise concerns for certain real-world applications. However, due to the overall lightweight design of TQNet and advancements in modern GPU parallelization technologies, the model can operate with near-linear computational overhead across varying numbers of channels. As illustrated in Figure 7 Right, TQNet maintains minimal computational overhead even with different channel counts, closely approaching the efficiency of linear-based methods (i.e., DLinear). This indicates that in most practical scenarios (e.g., when the number of channels $C < 1000$), TQNet can operate with remarkably low computational costs, making it highly suitable for real-world applications.

## 5. Limitations

This paper presents an efficient and robust approach for modeling multivariate correlations in time series forecasting. However, TQNet still has several practical limitations worth noting:

- Similar to CycleNet (Lin et al., 2024b), TQNet heavily relies on the inherent periodicity of the data to determine the hyperparameter $W$. This dependency may limit its generalization to datasets without clear periodic patterns.

- When the underlying multivariate correlations in the data are weak or insignificant, enforcing strong multivariate modeling may introduce unnecessary complexity and even negatively impact performance.

- As shown in Figure 8 in Appendix B, the benefits of multivariate modeling diminish as the look-back window becomes sufficiently long. On the one hand, a longer look-back provides richer temporal information,

which can partially substitute for multivariate cues. On the other hand, longer input windows introduce more noise, increasing the risk of overfitting in correlation modeling.

We believe that addressing these challenges will further enhance the robustness and generalizability of TQNet, and we leave this for future exploration.

## 6. Conclusion

This paper presented TQNet, a simple yet efficient model for multivariate time series forecasting. At the core of TQNet lies the Temporal Query (TQ) technique, an innovative method that employs periodically shifted learnable parameters as queries in the attention mechanism to model global inter-variable correlations. The TQ technique enhances the model's ability to capture complex multivariate dependencies, offering improved robustness, interpretability, and adaptability. Extensive experiments on 12 challenging real-world datasets demonstrated the effectiveness of TQNet, achieving state-of-the-art performance across diverse scenarios. Remarkably, TQNet maintains high computational efficiency, comparable to linear-based methods, even when applied to high-dimensional datasets with nearly 1,000 variables. This balance between accuracy and efficiency underscores TQNet's potential as a practical and robust solution for real-world forecasting challenges.

## Acknowledgements

This work is supported by Guangdong Major Project of Basic and Applied Basic Research (2019B030302002), Guangdong Provincial Natural Science Foundation Project (2025A1515010113) and the Major Key Project of PCL, China under Grant PCL2023A09.

## Impact Statement

This paper presents work whose goal is to advance the field of Machine Learning. There are many potential societal consequences of our work, none which we feel must be specifically highlighted here.

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

# A. More Details of TQNet

---
**Algorithm 1** Overall Pseudocode of TQNet
---

**Require:** Historical look-back input $X_t \in \mathbb{R}^{C \times L}$, cycle index $t \mod W \in \mathbb{N}_0$
**Ensure:** Forecasting output $\bar{Y}_t \in \mathbb{R}^{C \times H}$
1: Initialize **learnable** parameters $\theta_{\text{TQ}} \in \mathbb{R}^{C \times W} \leftarrow 0$
2: **if** Instance Normalization is applied **then**
3:    $\mu, \sigma \leftarrow \text{Mean}(X_t), \text{STD}(X_t)$
4:    $X_t \leftarrow \frac{X_t - \mu}{\sqrt{\sigma^2 + \epsilon}}$
5: **end if**
6: $\text{index}_{\text{TQ}} \in \mathbb{N}_0^L \leftarrow (t \mod W + \text{Range}(L)) \mod W$
7: $\theta_{\text{TQ}}^{t,L} \in \mathbb{R}^{C \times L} \leftarrow \theta_{\text{TQ}}[:, \text{index}_{\text{TQ}}]$
8: $h_{\text{attn}} \in \mathbb{R}^{C \times L} \leftarrow \text{Multi-Head Attention}(Q = \theta_{\text{TQ}}^{t,L}, K = X_t, V = X_t) + X_t$
9: $h'_{\text{attn}} \in \mathbb{R}^{C \times d} \leftarrow \text{Linear}(h_{\text{attn}})$
10: $h_{\text{mlp}} \in \mathbb{R}^{C \times d} \leftarrow \text{Multi-Layer Perceptron}(h'_{\text{attn}}) + h'_{\text{attn}}$
11: $\bar{Y}_t \in \mathbb{R}^{C \times H} \leftarrow \text{Linear}(\text{Dropout}(h_{\text{mlp}}))$
12: **if** Instance Normalization is applied **then**
13:    $\bar{Y}_t \leftarrow \bar{Y}_t \times \sqrt{\sigma^2 + \epsilon} + \mu$
14: **end if**

---

## A.1. Method Details

The pseudocode of TQNet is presented in Algorithm 1. It takes as input the historical look-back sequence $X_t \in \mathbb{R}^{C \times L}$ and the cycle index $t \mod W \in \mathbb{N}_0$, and outputs the predicted sequence $\bar{Y}_t \in \mathbb{R}^{C \times H}$.

The algorithm begins by initializing the **learnable** parameters $\theta_{\text{TQ}} \in \mathbb{R}^{C \times W}$ to zeros. Here, $W$ is a hyperparameter that represents the prior cycle length of the dataset. During training, $\theta_{\text{TQ}}$ is optimized to capture the underlying inter-variable correlations adaptively.

In the input processing stage, an *optional* instance normalization layer can be applied to $X_t$. This step removes the sample's mean and variance to mitigate potential issues arising from distributional shifts and ensure that the input sequence is stationary. If instance normalization is applied, the corresponding de-normalization is performed on the model output to recover the original scale.

Next, the $\text{index}_{\text{TQ}} \in \mathbb{N}_0^L$ is computed, representing the indices in $\theta_{\text{TQ}}$ to extract the segment $\theta_{\text{TQ}}^{t,L}$ for the current sample. This is efficiently calculated as:

$$\text{index}_{\text{TQ}} = (t \mod W + \text{Range}(L)) \mod W, \quad (9)$$

where $\text{Range}(L)$ generates the array $[0, 1, \ldots, L-1]$. Using these indices, $\theta_{\text{TQ}}[:, \text{index}_{\text{TQ}}]$ is extracted to obtain $\theta_{\text{TQ}}^{t,L} \in \mathbb{R}^{C \times L}$, which serves as the TQ vectors for the sample.

In the multivariate dependency modeling stage, the TQ vectors are incorporated into the multi-head attention (MHA)

mechanism. Specifically, $\theta_{\text{TQ}}^{t,L}$ is used as the query ($Q$), while the input sequence $X_t$ serves as the key ($K$) and value ($V$). To stabilize training and enhance robustness, a residual connection is applied by adding $X_t$ to the output of the MHA, yielding $h_{\text{attn}} \in \mathbb{R}^{C \times L}$.

In the temporal dependency modeling stage, the dimensionality of $h_{\text{attn}}$ is first transformed from $\mathbb{R}^{C \times L}$ to $\mathbb{R}^{C \times d}$ via a linear layer, resulting in $h'_{\text{attn}}$. Then, a two-layer multi-layer perceptron (MLP) with GeLU activation is used to extract non-linear temporal dependencies. Similar to the attention mechanism, a residual connection is added to stabilize training. This process outputs $h_{\text{mlp}} \in \mathbb{R}^{C \times d}$.

In the output stage, a linear layer with Dropout maps the learned hidden representations $h_{\text{mlp}}$ to the desired forecasting horizon $H$, resulting in the final output $\bar{Y}_t \in \mathbb{R}^{C \times H}$. If instance normalization was applied during the input stage, de-normalization is performed here to restore the predictions to their original scale.

## A.2. Experimental Details

The complete experimental details can be found in our open-source code repository[2]. Specifically, the experiments in this paper are implemented using PyTorch (Paszke et al., 2019) and executed on a single NVIDIA GeForce RTX 4090 GPU with 24 GB memory. The models are trained using the Adam optimizer (Kingma & Ba, 2014) and optimized with the L2 loss function. The training-validation-test splits are consistent with prior works, such as iTransformer (Liu et al., 2024c) and TimesNet (Wu et al., 2023). Specifically, the data splits follow a 6:2:2 ratio for the ETT and PEMS series datasets and a 7:1:2 ratio for the remaining datasets.

TQNet is trained for 30 epochs with early stopping based on a patience of 5 on the validation set. The learning rate is set to $3 \times 10^{-3}$ for most datasets, except for smaller datasets (i.e., the ETT series), where it is reduced to $1 \times 10^{-3}$. The batch size varies based on the dataset's scale to maximize GPU utilization while avoiding out-of-memory errors. For instance, a batch size of 16 is used for the Traffic dataset, while 64 is used for the Weather dataset. In TQNet, the number of attention heads in the multi-head attention mechanism is fixed at 4, and the dropout rate is set to 0.5 by default. Additionally, the dropout rate in the output layer is set to 0.5 for smaller datasets (e.g., the ETT series and Weather) and 0 for larger datasets. To ensure reproducibility, all experiments are conducted with a fixed random seed of 2024.

---

[2]https://github.com/ACAT-SCUT/TQNet

## A.3. Discussion about Hyperparameter $W$

The hyperparameter $W$, which represents the prior cycle length of the dataset, is determined based on domain-specific knowledge of periodic patterns and sampling intervals, as suggested by CycleNet (Lin et al., 2024b), or alternatively through computational techniques such as the autocorrelation function (ACF) (Madsen, 2007). The selected $W$ values for TQNet are aligned with the cycle length settings used in CycleNet, as summarized in Table 1.

For a detailed explanation of ACF and its use in identifying periodicities in time series data, we strongly refer readers to Appendix B.2 of the CycleNet paper (Lin et al., 2024b). We have provided the corresponding runnable code snippets in our open-source repository[3], and omit further details here for brevity.

Moreover, since TQNet considers only a single periodic length $W$ per dataset, it may encounter challenges when dealing with multi-period scenarios (e.g., hourly, daily, weekly, or monthly patterns). This issue can be discussed in two representative cases:

1. **Overlapping periodicities.** This scenario is generally manageable. For example, the Traffic dataset exhibits both daily (24-hour) and weekly (7×24-hour) patterns. In such cases, selecting the longest periodicity (weekly) is typically sufficient, as it implicitly captures the shorter cycle. In fact, most real-world datasets fall into this category, meaning that TQNet remains effective in practical applications.

2. **Irregularly interwoven periodicities.** In contrast, datasets with irregularly overlapping periodicities, such as weekly (7×24-hour) and monthly (30×24-hour) patterns, pose a greater challenge. Simply choosing the longest periodicity may fail to capture the nuances of the overlapping weekly cycle. A practical compromise in such cases is to focus solely on the daily periodicity (24-hour). Alternatively, one could ensemble multiple TQNet models, each configured with a different $W$, to improve adaptability to complex periodic structures. Overall, effectively modeling multi-periodicity remains an open research question, even for existing methods.

## A.4. Effectiveness analysis

The TQ technique lies at the core of TQNet, aiming to enhance the robustness of global inter-variable correlation modeling within the attention mechanism. In this subsection, we provide theoretical insights into why the TQ technique facilitates more stable and expressive representations of variable correlations.

---

[3]acf_plot.ipynb in https://github.com/ACAT-SCUT/TQNet

In TQ-MHA, the queries $Q$ are generated from globally shared, periodically shifted learnable vectors, while the keys $K$ and values $V$ are directly derived from the raw input sequence $X_t$, thereby encoding more localized, sample-specific patterns. For simplicity, we assume only a single attention head in MHA. During training, the model is optimized to maximize the attention output:

$$O = \text{Softmax}\left(\frac{Q^i K^{i\top}}{\sqrt{L}}\right) V^i, \qquad (10)$$

which implies that, for each input sample $i$, the learnable queries $Q^i$ is encouraged to align with the corresponding key $K^i$. This alignment allows the queries to capture more relevant inter-variable information, which is crucial for accurate forecasting. This alignment can be interpreted as maximizing their directional similarity:

$$\text{Corr}(Q^i) \approx \text{Corr}(K^i), \qquad (11)$$

where $\text{Corr}(\cdot) \in \left\{A \in [-1,1]^{C \times C} \mid A = A^\top, \text{diag}(A) = \mathbf{1}\right\}$ denotes a normalized correlation measure between the vectors.

Furthermore, since the query $Q^i$ is extracted periodically from the shared learnable parameter $\theta_{\text{TQ}}$, multiple samples spaced $W$ time steps apart will share the same queries. As a result, the effective correlation learned by $Q^i$ remains consistent across periods and is effectively an average over multiple local samples:

$$\text{Corr}(Q^i) = \text{Corr}(Q^{i+nW}), \quad n = 0, 1, \ldots, N-1, \quad (12)$$

and

$$\text{Corr}(Q^i) \approx \frac{1}{N} \sum_{n=0}^{N-1} \text{Corr}(K^{i+nW}), \qquad (13)$$

where $W$ is the periodic length, $N$ is the number of sampled periods, and $K^{i+nW}$ denotes the keys derived from the raw inputs at those respective time steps.

Therefore, in practice, the TQ-based queries $Q^i$ serves as an averaged representation of correlations across multiple periodic samples in the dataset. This averaging effect mitigates the influence of local non-stationary noise or outliers, leading to more stable and robust inter-variable correlation modeling.

Table 5: Full multivariate time series forecasting results of Table 2 for all prediction horizons $H \in \{96, 192, 336, 720\}$. The look-back length $L$ is fixed at 96, and the reproduced baseline results are sourced from TimeXer (Wang et al., 2024c), iTransformer (Liu et al., 2024c), and CycleNet (Lin et al., 2024b). The best results are highlighted in **bold**, while the second-best results are underlined.

| Model | | TQNet (Ours) | | TimeXer (2024c) | | CycleNet (2024b) | | iTransformer (2024c) | | MSGNet (2024) | | TimesNet (2023) | | PatchTST (2023) | | Crossformer (2023) | | DLinear (2023) | | SCINet (2022a) | |
|---|---|---|---|---|---|---|---|---|---|---|---|---|---|---|---|---|---|---|---|---|---|
| Metric | | MSE | MAE | MSE | MAE | MSE | MAE | MSE | MAE | MSE | MAE | MSE | MAE | MSE | MAE | MSE | MAE | MSE | MAE | MSE | MAE |
| ETTh1 | 96 | **0.371** | **0.393** | 0.382 | 0.403 | 0.375 | 0.395 | 0.386 | 0.405 | 0.390 | 0.411 | 0.384 | 0.402 | 0.414 | 0.419 | 0.423 | 0.448 | 0.386 | 0.400 | 0.654 | 0.599 |
| | 192 | **0.428** | **0.426** | 0.429 | 0.435 | 0.436 | 0.428 | 0.441 | 0.436 | 0.443 | 0.442 | 0.436 | 0.429 | 0.460 | 0.445 | 0.471 | 0.474 | 0.437 | 0.432 | 0.719 | 0.631 |
| | 336 | 0.476 | 0.446 | **0.468** | 0.448 | 0.496 | 0.455 | 0.487 | 0.458 | 0.482 | 0.469 | 0.491 | 0.469 | 0.501 | 0.466 | 0.570 | 0.546 | 0.481 | 0.459 | 0.778 | 0.659 |
| | 720 | 0.487 | 0.470 | 0.469 | 0.461 | 0.520 | 0.484 | 0.503 | 0.491 | 0.496 | 0.488 | 0.521 | 0.500 | 0.500 | 0.488 | 0.653 | 0.621 | 0.519 | 0.516 | 0.836 | 0.699 |
| | Avg | 0.441 | **0.434** | **0.437** | 0.437 | 0.457 | 0.441 | 0.454 | 0.448 | 0.453 | 0.453 | 0.458 | 0.450 | 0.469 | 0.455 | 0.529 | 0.522 | 0.456 | 0.452 | 0.747 | 0.647 |
| ETTh2 | 96 | 0.295 | 0.343 | **0.286** | **0.338** | 0.298 | 0.344 | 0.297 | 0.349 | 0.329 | 0.371 | 0.340 | 0.374 | 0.302 | 0.348 | 0.745 | 0.584 | 0.333 | 0.387 | 0.707 | 0.621 |
| | 192 | 0.367 | 0.393 | **0.363** | **0.389** | 0.372 | 0.396 | 0.380 | 0.400 | 0.402 | 0.414 | 0.402 | 0.414 | 0.388 | 0.400 | 0.877 | 0.656 | 0.477 | 0.476 | 0.860 | 0.689 |
| | 336 | 0.417 | 0.427 | **0.414** | **0.423** | 0.431 | 0.439 | 0.428 | 0.432 | 0.440 | 0.445 | 0.452 | 0.452 | 0.426 | 0.433 | 1.043 | 0.731 | 0.594 | 0.541 | 1.000 | 0.744 |
| | 720 | 0.433 | 0.446 | **0.408** | **0.432** | 0.450 | 0.458 | 0.427 | 0.445 | 0.480 | 0.477 | 0.462 | 0.468 | 0.431 | 0.446 | 1.104 | 0.763 | 0.831 | 0.657 | 1.249 | 0.838 |
| | Avg | 0.378 | 0.402 | **0.368** | **0.396** | 0.388 | 0.409 | 0.383 | 0.407 | 0.413 | 0.427 | 0.414 | 0.427 | 0.387 | 0.407 | 0.942 | 0.684 | 0.559 | 0.515 | 0.954 | 0.723 |
| ETTm1 | 96 | **0.311** | **0.353** | 0.318 | 0.356 | 0.319 | 0.360 | 0.334 | 0.368 | 0.319 | 0.366 | 0.338 | 0.375 | 0.329 | 0.367 | 0.404 | 0.426 | 0.345 | 0.372 | 0.418 | 0.438 |
| | 192 | **0.356** | **0.378** | 0.362 | 0.383 | 0.360 | 0.381 | 0.377 | 0.391 | 0.377 | 0.397 | 0.374 | 0.387 | 0.367 | 0.385 | 0.450 | 0.451 | 0.380 | 0.389 | 0.439 | 0.450 |
| | 336 | 0.390 | 0.401 | 0.395 | 0.407 | **0.389** | 0.403 | 0.426 | 0.420 | 0.417 | 0.422 | 0.410 | 0.411 | 0.399 | 0.410 | 0.532 | 0.515 | 0.413 | 0.413 | 0.490 | 0.485 |
| | 720 | 0.452 | 0.440 | 0.452 | 0.441 | **0.447** | 0.441 | 0.491 | 0.459 | 0.487 | 0.463 | 0.478 | 0.450 | 0.454 | **0.439** | 0.666 | 0.589 | 0.474 | 0.453 | 0.595 | 0.550 |
| | Avg | **0.377** | **0.393** | 0.382 | 0.397 | 0.379 | 0.396 | 0.407 | 0.410 | 0.400 | 0.412 | 0.400 | 0.406 | 0.387 | 0.400 | 0.513 | 0.495 | 0.403 | 0.407 | 0.486 | 0.481 |
| ETTm2 | 96 | 0.173 | 0.256 | 0.171 | 0.256 | **0.163** | **0.246** | 0.180 | 0.264 | 0.182 | 0.266 | 0.187 | 0.267 | 0.175 | 0.259 | 0.287 | 0.366 | 0.193 | 0.292 | 0.286 | 0.377 |
| | 192 | 0.238 | 0.298 | 0.237 | 0.299 | **0.229** | **0.290** | 0.250 | 0.309 | 0.248 | 0.306 | 0.249 | 0.309 | 0.241 | 0.302 | 0.414 | 0.492 | 0.284 | 0.362 | 0.399 | 0.445 |
| | 336 | 0.301 | 0.340 | 0.296 | 0.338 | **0.284** | **0.327** | 0.311 | 0.348 | 0.312 | 0.346 | 0.321 | 0.351 | 0.305 | 0.343 | 0.597 | 0.542 | 0.369 | 0.427 | 0.637 | 0.591 |
| | 720 | 0.397 | 0.396 | 0.392 | 0.394 | **0.389** | **0.391** | 0.412 | 0.407 | 0.414 | 0.404 | 0.408 | 0.403 | 0.402 | 0.400 | 1.730 | 1.042 | 0.554 | 0.522 | 0.960 | 0.735 |
| | Avg | 0.277 | 0.323 | 0.274 | 0.322 | **0.266** | **0.314** | 0.288 | 0.332 | 0.289 | 0.330 | 0.291 | 0.333 | 0.281 | 0.326 | 0.757 | 0.611 | 0.350 | 0.401 | 0.571 | 0.537 |
| Electricity | 96 | **0.134** | 0.229 | 0.140 | 0.242 | 0.136 | **0.229** | 0.148 | 0.240 | 0.165 | 0.274 | 0.168 | 0.272 | 0.181 | 0.270 | 0.219 | 0.314 | 0.197 | 0.282 | 0.286 | 0.377 |
| | 192 | 0.154 | 0.247 | 0.157 | 0.256 | **0.152** | **0.244** | 0.162 | 0.253 | 0.185 | 0.292 | 0.184 | 0.289 | 0.188 | 0.274 | 0.231 | 0.322 | 0.196 | 0.285 | 0.399 | 0.445 |
| | 336 | **0.169** | 0.264 | 0.176 | 0.275 | 0.170 | **0.264** | 0.178 | 0.269 | 0.197 | 0.304 | 0.198 | 0.300 | 0.204 | 0.293 | 0.246 | 0.337 | 0.209 | 0.301 | 0.637 | 0.591 |
| | 720 | **0.201** | **0.294** | 0.211 | 0.306 | 0.212 | 0.299 | 0.225 | 0.317 | 0.231 | 0.332 | 0.220 | 0.320 | 0.246 | 0.324 | 0.280 | 0.363 | 0.245 | 0.333 | 0.960 | 0.735 |
| | Avg | **0.164** | **0.259** | 0.171 | 0.270 | 0.168 | 0.259 | 0.178 | 0.270 | 0.194 | 0.301 | 0.193 | 0.295 | 0.205 | 0.290 | 0.244 | 0.334 | 0.212 | 0.300 | 0.571 | 0.537 |
| Solar-Energy | 96 | **0.173** | **0.233** | 0.215 | 0.295 | 0.190 | 0.247 | 0.203 | 0.237 | 0.210 | 0.246 | 0.250 | 0.292 | 0.234 | 0.286 | 0.310 | 0.331 | 0.290 | 0.378 | 0.237 | 0.344 |
| | 192 | **0.199** | **0.257** | 0.236 | 0.301 | 0.210 | 0.266 | 0.233 | 0.261 | 0.265 | 0.290 | 0.296 | 0.318 | 0.267 | 0.310 | 0.734 | 0.725 | 0.320 | 0.398 | 0.280 | 0.380 |
| | 336 | **0.211** | **0.263** | 0.252 | 0.307 | 0.217 | 0.266 | 0.248 | 0.273 | 0.294 | 0.318 | 0.319 | 0.330 | 0.290 | 0.315 | 0.750 | 0.735 | 0.353 | 0.415 | 0.304 | 0.389 |
| | 720 | **0.209** | 0.270 | 0.244 | 0.305 | 0.223 | 0.266 | 0.249 | 0.275 | 0.285 | 0.315 | 0.338 | 0.337 | 0.289 | 0.317 | 0.769 | 0.765 | 0.356 | 0.413 | 0.308 | 0.388 |
| | Avg | **0.198** | **0.256** | 0.237 | 0.302 | 0.210 | 0.261 | 0.233 | 0.262 | 0.263 | 0.292 | 0.301 | 0.319 | 0.270 | 0.307 | 0.641 | 0.639 | 0.330 | 0.401 | 0.282 | 0.375 |
| Traffic | 96 | 0.413 | **0.261** | 0.428 | 0.271 | 0.458 | 0.296 | **0.395** | 0.268 | 0.608 | 0.349 | 0.593 | 0.321 | 0.462 | 0.290 | 0.522 | 0.290 | 0.650 | 0.396 | 0.788 | 0.499 |
| | 192 | 0.432 | **0.271** | 0.448 | 0.282 | 0.457 | 0.294 | **0.417** | 0.276 | 0.634 | 0.371 | 0.617 | 0.336 | 0.466 | 0.290 | 0.530 | 0.293 | 0.598 | 0.370 | 0.789 | 0.505 |
| | 336 | 0.450 | **0.277** | 0.473 | 0.289 | 0.470 | 0.299 | **0.433** | 0.283 | 0.669 | 0.388 | 0.629 | 0.336 | 0.482 | 0.300 | 0.558 | 0.305 | 0.605 | 0.373 | 0.797 | 0.508 |
| | 720 | 0.486 | **0.295** | 0.516 | 0.307 | 0.502 | 0.314 | **0.467** | 0.302 | 0.729 | 0.420 | 0.640 | 0.350 | 0.514 | 0.320 | 0.589 | 0.328 | 0.645 | 0.394 | 0.841 | 0.523 |
| | Avg | 0.445 | **0.276** | 0.466 | 0.287 | 0.472 | 0.301 | **0.428** | 0.282 | 0.660 | 0.382 | 0.620 | 0.336 | 0.481 | 0.300 | 0.550 | 0.304 | 0.625 | 0.383 | 0.804 | 0.509 |
| Weather | 96 | **0.157** | **0.200** | 0.157 | 0.205 | 0.158 | 0.203 | 0.174 | 0.214 | 0.163 | 0.212 | 0.172 | 0.220 | 0.177 | 0.210 | 0.158 | 0.230 | 0.196 | 0.255 | 0.221 | 0.306 |
| | 192 | 0.206 | **0.245** | **0.204** | 0.247 | 0.207 | 0.247 | 0.221 | 0.254 | 0.211 | 0.254 | 0.219 | 0.261 | 0.225 | 0.250 | 0.206 | 0.277 | 0.237 | 0.296 | 0.261 | 0.340 |
| | 336 | 0.262 | **0.287** | **0.261** | 0.290 | 0.262 | 0.289 | 0.278 | 0.296 | 0.273 | 0.299 | 0.280 | 0.306 | 0.278 | 0.290 | 0.272 | 0.335 | 0.283 | 0.335 | 0.309 | 0.378 |
| | 720 | 0.344 | 0.342 | **0.340** | 0.341 | 0.344 | 0.344 | 0.358 | 0.349 | 0.351 | 0.348 | 0.365 | 0.359 | 0.354 | **0.340** | 0.398 | 0.418 | 0.345 | 0.381 | 0.377 | 0.427 |
| | Avg | 0.242 | **0.269** | **0.241** | 0.271 | 0.243 | 0.271 | 0.258 | 0.278 | 0.249 | 0.278 | 0.259 | 0.287 | 0.259 | 0.273 | 0.259 | 0.315 | 0.265 | 0.317 | 0.292 | 0.363 |
| PEMS03 | 96 | **0.060** | **0.161** | 0.070 | 0.173 | 0.066 | 0.172 | 0.071 | 0.174 | 0.078 | 0.187 | 0.085 | 0.192 | 0.099 | 0.216 | 0.090 | 0.203 | 0.122 | 0.243 | 0.066 | 0.172 |
| | 192 | **0.077** | **0.182** | 0.092 | 0.194 | 0.089 | 0.201 | 0.093 | 0.201 | 0.108 | 0.218 | 0.118 | 0.223 | 0.142 | 0.259 | 0.121 | 0.240 | 0.201 | 0.317 | 0.085 | 0.198 |
| | 336 | **0.104** | **0.215** | 0.129 | 0.229 | 0.136 | 0.247 | 0.125 | 0.236 | 0.178 | 0.272 | 0.155 | 0.260 | 0.211 | 0.319 | 0.202 | 0.317 | 0.333 | 0.425 | 0.127 | 0.238 |
| | 720 | **0.148** | **0.253** | 0.157 | 0.261 | 0.182 | 0.282 | 0.164 | 0.275 | 0.238 | 0.328 | 0.228 | 0.317 | 0.269 | 0.370 | 0.262 | 0.367 | 0.457 | 0.515 | 0.178 | 0.287 |
| | Avg | **0.097** | **0.203** | 0.112 | 0.214 | 0.118 | 0.226 | 0.113 | 0.222 | 0.150 | 0.251 | 0.147 | 0.248 | 0.180 | 0.291 | 0.169 | 0.282 | 0.278 | 0.375 | 0.114 | 0.224 |
| PEMS04 | 96 | **0.067** | **0.166** | 0.074 | 0.178 | 0.078 | 0.186 | 0.078 | 0.183 | 0.086 | 0.199 | 0.087 | 0.195 | 0.105 | 0.224 | 0.098 | 0.218 | 0.148 | 0.272 | 0.073 | 0.177 |
| | 192 | **0.077** | **0.181** | 0.087 | 0.195 | 0.099 | 0.212 | 0.095 | 0.205 | 0.101 | 0.218 | 0.103 | 0.215 | 0.153 | 0.275 | 0.131 | 0.256 | 0.224 | 0.340 | 0.084 | 0.193 |
| | 336 | **0.097** | **0.206** | 0.110 | 0.214 | 0.133 | 0.248 | 0.120 | 0.233 | 0.127 | 0.247 | 0.136 | 0.250 | 0.229 | 0.339 | 0.205 | 0.326 | 0.355 | 0.437 | 0.099 | 0.211 |
| | 720 | 0.123 | 0.233 | 0.148 | 0.251 | 0.167 | 0.281 | 0.150 | 0.262 | 0.174 | 0.292 | 0.190 | 0.303 | 0.291 | 0.389 | 0.402 | 0.457 | 0.452 | 0.504 | **0.114** | **0.227** |
| | Avg | **0.091** | **0.197** | 0.105 | 0.209 | 0.119 | 0.232 | 0.111 | 0.221 | 0.122 | 0.239 | 0.129 | 0.241 | 0.195 | 0.307 | 0.209 | 0.314 | 0.295 | 0.388 | 0.093 | 0.202 |
| PEMS07 | 96 | **0.051** | **0.143** | 0.057 | 0.152 | 0.062 | 0.162 | 0.067 | 0.165 | 0.079 | 0.182 | 0.082 | 0.181 | 0.095 | 0.207 | 0.094 | 0.200 | 0.115 | 0.242 | 0.068 | 0.171 |
| | 192 | **0.063** | **0.159** | 0.079 | 0.179 | 0.086 | 0.192 | 0.088 | 0.190 | 0.099 | 0.206 | 0.101 | 0.204 | 0.150 | 0.262 | 0.139 | 0.247 | 0.210 | 0.329 | 0.119 | 0.225 |
| | 336 | **0.081** | **0.179** | 0.099 | 0.191 | 0.128 | 0.234 | 0.110 | 0.215 | 0.133 | 0.239 | 0.134 | 0.238 | 0.253 | 0.340 | 0.311 | 0.369 | 0.398 | 0.458 | 0.149 | 0.237 |
| | 720 | **0.103** | **0.203** | 0.107 | 0.205 | 0.176 | 0.268 | 0.139 | 0.245 | 0.179 | 0.279 | 0.181 | 0.279 | 0.346 | 0.404 | 0.396 | 0.442 | 0.594 | 0.553 | 0.141 | 0.234 |
| | Avg | **0.075** | **0.171** | 0.085 | 0.182 | 0.113 | 0.214 | 0.101 | 0.204 | 0.122 | 0.227 | 0.125 | 0.226 | 0.211 | 0.303 | 0.235 | 0.315 | 0.329 | 0.396 | 0.119 | 0.217 |
| PEMS08 | 96 | **0.071** | **0.170** | 0.075 | 0.176 | 0.082 | 0.185 | 0.079 | 0.182 | 0.105 | 0.211 | 0.112 | 0.212 | 0.168 | 0.232 | 0.165 | 0.214 | 0.154 | 0.276 | 0.087 | 0.184 |
| | 192 | **0.096** | **0.196** | 0.102 | 0.201 | 0.117 | 0.226 | 0.115 | 0.219 | 0.141 | 0.243 | 0.141 | 0.238 | 0.224 | 0.281 | 0.215 | 0.260 | 0.248 | 0.353 | 0.122 | 0.221 |
| | 336 | **0.149** | **0.244** | 0.158 | 0.248 | 0.169 | 0.268 | 0.186 | 0.235 | 0.211 | 0.300 | 0.198 | 0.283 | 0.321 | 0.354 | 0.315 | 0.355 | 0.440 | 0.470 | 0.189 | 0.270 |
| | 720 | 0.253 | 0.309 | 0.366 | 0.377 | 0.233 | 0.306 | **0.221** | **0.267** | 0.364 | 0.387 | 0.320 | 0.351 | 0.408 | 0.417 | 0.377 | 0.397 | 0.674 | 0.565 | 0.236 | 0.300 |
| | Avg | **0.142** | 0.229 | 0.175 | 0.250 | 0.150 | 0.246 | 0.150 | **0.226** | 0.205 | 0.285 | 0.193 | 0.271 | 0.280 | 0.321 | 0.268 | 0.307 | 0.379 | 0.416 | 0.159 | 0.244 |

# B. More Results of TQNet

## B.1. Full Comparison Results

Table 5 presents the full comparison results of TQNet against several baseline methods across 12 real-world multivariate datasets. The results demonstrate that TQNet consistently achieves state-of-the-art forecasting performance under most experimental settings, underscoring the effectiveness of the proposed approach.

## B.2. Univariate Forecasting Results

Previously, we primarily demonstrated the performance of TQNet in multivariate forecasting scenarios. However, in real-world applications, a more common setting is multivariate-to-univariate forecasting, where exogenous variables are utilized to predict a single target variable. Therefore, we further present a comparison of forecasting results under this setting (see Table 6). As shown, TQNet exhibits strong competitiveness compared to state-of-the-art models specifically designed for multivariate-to-univariate tasks, such as TimeXer.

## B.3. Impact of Module Stacking

TQNet achieves state-of-the-art performance using only the most essential components: a single-layer attention mechanism and a single MLP. This design achieves an effective balance between forecasting accuracy and computational efficiency. To investigate whether increasing model capacity leads to further improvements, we experiment with stacking three layers of TQ-MHA and MLP modules within TQNet.

The results in Figure 7 show that adding more stacked modules does not yield significant performance gains. In most datasets, performance slightly declines, although modest improvements are observed on the PEMS datasets. This outcome highlights the robustness and soundness of the original TQNet design, which already delivers near-optimal performance with minimal architectural complexity.

These findings not only validate the effectiveness of our approach but also reinforce our central claim: TQNet strikes an ideal trade-off between forecasting accuracy and computational cost. Finally, we also advocate for lightweight model designs in time series forecasting, as they enhance interpretability, facilitate practical deployment, and maintain strong predictive performance.

## B.4. Comparison with ETS

ETS (Exponential Smoothing (Hyndman & Athanasopoulos, 2018)) is a classical forecasting method that leverages periodic patterns in time series data. In Table 8, we compare the performance of TQNet and ETS across multiple datasets, demonstrating the superior forecasting capabilities

of TQNet. The following points are worth noting when interpreting the results:

1. ETS is fundamentally a univariate forecasting method and cannot leverage multivariate information, whereas this is a key capability emphasized by TQNet. This limitation puts ETS at a disadvantage in case where inter-variable dependencies play an important role.

2. ETS requires the look-back window $L$ to satisfy $L \geq 2W$, where $W$ denotes the period length. Accordingly, we set $L = 336$ and $L = 720$ for ETS to meet this constraint, instead of using the default $L = 96$ adopted in the TQNet experiments.

3. ETS is primarily designed for short-term forecasting based on statistical extrapolation. When applied to long-horizon forecasting, its predictions often diverge significantly from the ground truth, with errors compounding over time. This explains the substantial increase in MSE observed for ETS under large forecasting horizons (e.g., $H = 720$).

## B.5. Impact of Look-back Length

Table 8 illustrates the impact of varying look-back lengths on the performance of TQNet and other models. The results reveal that under shorter input lengths, CD-based methods (e.g., iTransformer) exhibit a significant advantage over CI-based methods (e.g., PatchTST). In these scenarios, TQNet outperforms other CD-based methods, demonstrating the effectiveness of the proposed TQ technique in capturing robust multivariate dependencies.

When the input length increases, models can leverage more temporal dependencies to compensate for the limitations in modeling multivariate relationships. Consequently, the performance gap between CD-based and CI-based methods diminishes. However, overall, CD-based methods maintain a distinct advantage and are more suitable for practical deployment. For instance, in real-world scenarios where it may be challenging to obtain sufficiently long historical data, CD-based methods can effectively enhance predictive performance by leveraging superior multivariate dependency modeling.

## B.6. Robustness of TQNet

To evaluate the robustness of TQNet, we conducted multiple runs of the model under different random seeds and learning rates. The results indicate that TQNet consistently maintains low standard deviations across all settings, demonstrating its stability and reliability. This robustness suggests that TQNet is resilient to variations in initialization and hyperparameter configurations, making it a dependable choice for multivariate time series forecasting tasks.

Table 6: Multivariate-to-univariate forecasting time series forecasting results. The look-back length $L$ is fixed at 96, and the reproduced baseline results are sourced from TimeXer (Wang et al., 2024c). The best results are highlighted in **bold**, while the second-best results are underlined.

| Model | | TQNet (Ours) | | TimeXer (2024c) | | iTransformer (2024c) | | TimesNet (2023) | | PatchTST (2023) | | Crossformer (2023) | | DLinear (2023) | | SCINet (2022a) | | NSformer (2022b) | | Autoformer (2021) | |
|---|---|---|---|---|---|---|---|---|---|---|---|---|---|---|---|---|---|---|---|---|---|
| Metric | | MSE | MAE | MSE | MAE | MSE | MAE | MSE | MAE | MSE | MAE | MSE | MAE | MSE | MAE | MSE | MAE | MSE | MAE | MSE | MAE |
| ETTm2 | 96 | **0.064** | **0.181** | 0.067 | 0.188 | 0.071 | 0.194 | 0.073 | 0.200 | 0.068 | 0.188 | 0.149 | 0.309 | 0.072 | 0.195 | 0.253 | 0.427 | 0.098 | 0.229 | 0.133 | 0.282 |
| | 192 | **0.097** | **0.231** | 0.101 | 0.236 | 0.108 | 0.247 | 0.106 | 0.247 | 0.100 | 0.236 | 0.686 | 0.740 | 0.105 | 0.240 | 0.592 | 0.677 | 0.161 | 0.302 | 0.143 | 0.294 |
| | 336 | 0.128 | 0.272 | 0.130 | 0.275 | 0.140 | 0.288 | 0.150 | 0.296 | **0.128** | **0.271** | 0.546 | 0.602 | 0.136 | 0.280 | 0.777 | 0.790 | 0.243 | 0.362 | 0.156 | 0.308 |
| | 720 | **0.181** | **0.330** | 0.182 | 0.332 | 0.188 | 0.340 | 0.186 | 0.338 | 0.185 | 0.335 | 2.524 | 1.424 | 0.191 | 0.335 | 1.117 | 0.960 | 0.326 | 0.441 | 0.184 | 0.333 |
| | Avg | **0.118** | **0.254** | 0.120 | 0.258 | 0.127 | 0.267 | 0.129 | 0.270 | 0.120 | 0.258 | 0.976 | 0.769 | 0.126 | 0.263 | 0.685 | 0.714 | 0.207 | 0.334 | 0.154 | 0.304 |
| Electricity | 96 | **0.239** | **0.348** | 0.261 | 0.366 | 0.299 | 0.403 | 0.342 | 0.437 | 0.339 | 0.412 | 0.265 | 0.364 | 0.387 | 0.451 | 0.390 | 0.462 | 0.298 | 0.407 | 0.432 | 0.502 |
| | 192 | **0.283** | **0.375** | 0.316 | 0.397 | 0.321 | 0.413 | 0.384 | 0.461 | 0.361 | 0.425 | 0.313 | 0.390 | 0.365 | 0.436 | 0.375 | 0.456 | 0.340 | 0.433 | 0.492 | 0.492 |
| | 336 | **0.342** | **0.415** | 0.367 | 0.429 | 0.379 | 0.446 | 0.439 | 0.493 | 0.393 | 0.440 | 0.380 | 0.431 | 0.391 | 0.453 | 0.468 | 0.519 | 0.405 | 0.471 | 0.508 | 0.548 |
| | 720 | 0.427 | 0.477 | 0.365 | **0.439** | 0.461 | 0.504 | 0.473 | 0.514 | 0.482 | 0.507 | 0.418 | 0.463 | 0.428 | 0.487 | 0.477 | 0.524 | 0.444 | 0.489 | 0.547 | 0.569 |
| | Avg | **0.323** | **0.404** | 0.327 | 0.408 | 0.365 | 0.442 | 0.410 | 0.476 | 0.394 | 0.446 | 0.344 | 0.412 | 0.393 | 0.457 | 0.428 | 0.490 | 0.372 | 0.450 | 0.495 | 0.528 |
| Traffic | 96 | **0.129** | **0.201** | 0.151 | 0.224 | 0.156 | 0.236 | 0.154 | 0.249 | 0.176 | 0.253 | 0.154 | 0.230 | 0.268 | 0.351 | 0.371 | 0.448 | 0.214 | 0.323 | 0.290 | 0.290 |
| | 192 | **0.131** | **0.204** | 0.152 | 0.229 | 0.156 | 0.237 | 0.164 | 0.255 | 0.162 | 0.243 | 0.180 | 0.256 | 0.302 | 0.387 | 0.450 | 0.503 | 0.195 | 0.307 | 0.291 | 0.291 |
| | 336 | **0.131** | **0.208** | 0.150 | 0.232 | 0.154 | 0.243 | 0.167 | 0.259 | 0.164 | 0.248 | 0.164 | 0.241 | 0.298 | 0.384 | 0.447 | 0.501 | 0.198 | 0.309 | 0.322 | 0.416 |
| | 720 | **0.148** | **0.228** | 0.172 | 0.253 | 0.177 | 0.268 | 0.197 | 0.292 | 0.189 | 0.267 | 0.203 | 0.277 | 0.340 | 0.416 | 0.521 | 0.548 | 0.835 | 0.507 | 0.307 | 0.414 |
| | Avg | **0.135** | **0.210** | 0.156 | 0.235 | 0.161 | 0.246 | 0.171 | 0.264 | 0.173 | 0.253 | 0.175 | 0.251 | 0.302 | 0.385 | 0.447 | 0.500 | 0.361 | 0.362 | 0.303 | 0.353 |

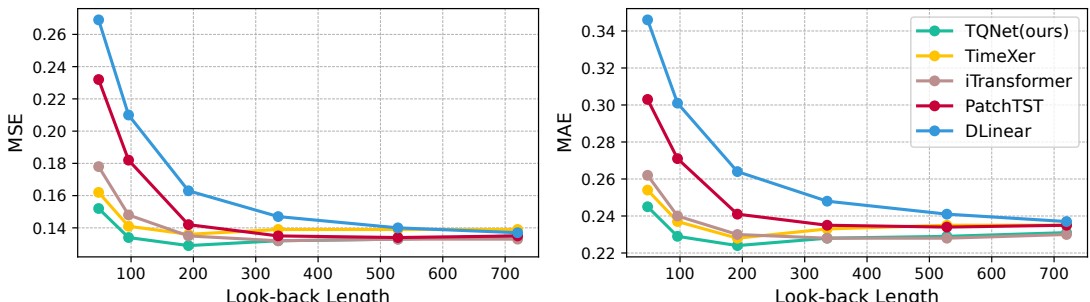

Figure 8: Performance of TQNet and comparative models on the Electricity dataset with different look-back lengths. The forecast horizon is set as 96.

Table 7: Comparison of the performance between the original TQNet model and the stacked version (three layers) on multiple datasets. Results are averaged across forecasting horizons $H \in \{96, 192, 336, 720\}$.

| Model | Original | | Stacking | |
|---|---|---|---|---|
| Metric | MSE | MAE | MSE | MAE |
| ETTh1 | **0.441** | **0.434** | 0.443 | 0.447 |
| ETTh2 | **0.378** | **0.402** | 0.382 | 0.405 |
| ETTm1 | **0.377** | **0.393** | 0.391 | 0.404 |
| ETTm2 | **0.277** | **0.323** | 0.280 | 0.324 |
| Electricity | **0.164** | **0.259** | 0.167 | 0.262 |
| Solar | **0.198** | **0.256** | 0.203 | 0.265 |
| traffic | **0.445** | **0.276** | 0.451 | 0.285 |
| weather | **0.242** | **0.269** | 0.242 | 0.271 |
| PEMS03 | 0.097 | 0.203 | **0.095** | **0.197** |
| PEMS04 | 0.091 | 0.197 | **0.084** | **0.189** |
| PEMS07 | 0.075 | 0.171 | **0.072** | **0.167** |
| PEMS08 | **0.142** | 0.229 | 0.147 | **0.225** |

Table 8: Comparison of TQNet and ETS for multivariate-to-univariate forecasting across multiple datasets. Results are averaged across forecasting horizons $H \in \{96, 192, 336, 720\}$.

| Model | TQNet | | ETS | |
|---|---|---|---|---|
| Metric | MSE | MAE | MSE | MAE |
| ETTh1 | **0.074** | **0.213** | 0.206 | 0.337 |
| ETTh2 | **0.178** | **0.337** | 231.3 | 8.006 |
| ETTm1 | **0.048** | **0.164** | 0.169 | 0.268 |
| ETTm2 | **0.121** | **0.263** | 12.19 | 1.788 |
| Electricity | **0.299** | **0.388** | 0.778 | 0.599 |
| Solar | **0.260** | **0.299** | 18.06 | 1.160 |
| Traffic | **0.122** | **0.201** | 92.69 | 0.490 |
| Weather | **0.002** | **0.029** | 0.007 | 0.044 |
| PEMS03 | **0.151** | **0.245** | 0.618 | 0.357 |
| PEMS04 | **0.063** | **0.186** | 0.345 | 0.352 |
| PEMS07 | **0.094** | **0.233** | 0.253 | 0.359 |
| PEMS08 | **0.178** | **0.309** | 0.426 | 0.458 |

Table 9: Performance of TQNet under different random seeds and learning rates. *Mean* represents the average value, and *Std* denotes the standard deviation.

| Setup | | Random Seed | | | | | | | | | | Learning Rate | | | | | | | | | |
|---|---|---|---|---|---|---|---|---|---|---|---|---|---|---|---|---|---|---|---|---|---|
| | | 2024 | | 2025 | | 2026 | | Mean | | Std | | 0.001 | | 0.002 | | 0.003 | | Mean | | Std | |
| Metric | | MSE | MAE | MSE | MAE | MSE | MAE | MSE | MAE | MSE | MAE | MSE | MAE | MSE | MAE | MSE | MAE | MSE | MAE | MSE | MAE |
| ETTh1 | 96 | 0.371 | 0.393 | 0.371 | 0.394 | 0.372 | 0.393 | 0.371 | 0.393 | 0.001 | 0.000 | 0.371 | 0.393 | 0.376 | 0.398 | 0.378 | 0.396 | 0.375 | 0.395 | 0.004 | 0.002 |
| | 192 | 0.428 | 0.426 | 0.430 | 0.424 | 0.430 | 0.423 | 0.429 | 0.424 | 0.001 | 0.002 | 0.428 | 0.426 | 0.435 | 0.429 | 0.435 | 0.432 | 0.433 | 0.429 | 0.004 | 0.003 |
| | 336 | 0.476 | 0.446 | 0.481 | 0.453 | 0.476 | 0.446 | 0.478 | 0.448 | 0.003 | 0.004 | 0.476 | 0.446 | 0.480 | 0.448 | 0.494 | 0.458 | 0.483 | 0.451 | 0.010 | 0.007 |
| | 720 | 0.487 | 0.470 | 0.510 | 0.487 | 0.491 | 0.472 | 0.496 | 0.476 | 0.012 | 0.009 | 0.487 | 0.470 | 0.510 | 0.486 | 0.521 | 0.493 | 0.506 | 0.483 | 0.017 | 0.012 |
| ETTh2 | 96 | 0.295 | 0.343 | 0.295 | 0.346 | 0.295 | 0.343 | 0.295 | 0.344 | 0.000 | 0.002 | 0.295 | 0.343 | 0.302 | 0.346 | 0.301 | 0.348 | 0.299 | 0.346 | 0.004 | 0.002 |
| | 192 | 0.367 | 0.393 | 0.370 | 0.394 | 0.366 | 0.391 | 0.368 | 0.392 | 0.002 | 0.002 | 0.367 | 0.393 | 0.374 | 0.393 | 0.376 | 0.396 | 0.372 | 0.394 | 0.005 | 0.002 |
| | 336 | 0.417 | 0.427 | 0.418 | 0.427 | 0.415 | 0.428 | 0.417 | 0.428 | 0.001 | 0.001 | 0.417 | 0.427 | 0.418 | 0.433 | 0.424 | 0.434 | 0.419 | 0.431 | 0.004 | 0.004 |
| | 720 | 0.433 | 0.446 | 0.441 | 0.450 | 0.441 | 0.451 | 0.438 | 0.449 | 0.004 | 0.003 | 0.433 | 0.446 | 0.437 | 0.449 | 0.437 | 0.449 | 0.436 | 0.448 | 0.002 | 0.002 |
| ETTm1 | 96 | 0.311 | 0.353 | 0.316 | 0.355 | 0.310 | 0.352 | 0.312 | 0.353 | 0.003 | 0.001 | 0.311 | 0.353 | 0.315 | 0.356 | 0.321 | 0.359 | 0.316 | 0.356 | 0.005 | 0.003 |
| | 192 | 0.356 | 0.378 | 0.358 | 0.380 | 0.362 | 0.382 | 0.359 | 0.380 | 0.003 | 0.002 | 0.356 | 0.378 | 0.365 | 0.384 | 0.361 | 0.381 | 0.361 | 0.381 | 0.004 | 0.003 |
| | 336 | 0.390 | 0.401 | 0.389 | 0.402 | 0.392 | 0.402 | 0.390 | 0.402 | 0.001 | 0.001 | 0.390 | 0.401 | 0.398 | 0.407 | 0.393 | 0.405 | 0.393 | 0.404 | 0.004 | 0.003 |
| | 720 | 0.452 | 0.440 | 0.451 | 0.441 | 0.452 | 0.441 | 0.452 | 0.441 | 0.001 | 0.001 | 0.452 | 0.440 | 0.460 | 0.447 | 0.455 | 0.443 | 0.456 | 0.443 | 0.004 | 0.003 |
| ETTm2 | 96 | 0.173 | 0.256 | 0.172 | 0.255 | 0.174 | 0.255 | 0.173 | 0.255 | 0.001 | 0.001 | 0.173 | 0.256 | 0.174 | 0.256 | 0.175 | 0.258 | 0.174 | 0.256 | 0.001 | 0.001 |
| | 192 | 0.238 | 0.298 | 0.239 | 0.299 | 0.241 | 0.300 | 0.239 | 0.299 | 0.002 | 0.001 | 0.238 | 0.298 | 0.240 | 0.299 | 0.244 | 0.301 | 0.241 | 0.299 | 0.003 | 0.002 |
| | 336 | 0.301 | 0.340 | 0.305 | 0.342 | 0.298 | 0.338 | 0.301 | 0.340 | 0.003 | 0.002 | 0.301 | 0.340 | 0.306 | 0.342 | 0.302 | 0.340 | 0.303 | 0.341 | 0.003 | 0.001 |
| | 720 | 0.397 | 0.396 | 0.397 | 0.396 | 0.398 | 0.396 | 0.397 | 0.396 | 0.000 | 0.000 | 0.397 | 0.396 | 0.396 | 0.395 | 0.396 | 0.395 | 0.396 | 0.395 | 0.001 | 0.001 |
| Electricity | 96 | 0.134 | 0.229 | 0.135 | 0.230 | 0.135 | 0.229 | 0.134 | 0.230 | 0.000 | 0.000 | 0.136 | 0.231 | 0.135 | 0.230 | 0.135 | 0.230 | 0.135 | 0.230 | 0.001 | 0.000 |
| | 192 | 0.154 | 0.247 | 0.153 | 0.246 | 0.153 | 0.246 | 0.153 | 0.246 | 0.000 | 0.000 | 0.154 | 0.247 | 0.152 | 0.246 | 0.152 | 0.246 | 0.153 | 0.246 | 0.001 | 0.001 |
| | 336 | 0.169 | 0.264 | 0.168 | 0.264 | 0.169 | 0.264 | 0.169 | 0.264 | 0.000 | 0.000 | 0.169 | 0.263 | 0.169 | 0.264 | 0.168 | 0.263 | 0.169 | 0.263 | 0.001 | 0.000 |
| | 720 | 0.201 | 0.294 | 0.204 | 0.296 | 0.210 | 0.302 | 0.205 | 0.298 | 0.005 | 0.004 | 0.202 | 0.292 | 0.201 | 0.292 | 0.201 | 0.294 | 0.201 | 0.293 | 0.001 | 0.001 |
| PEMS03 | 12 | 0.060 | 0.161 | 0.061 | 0.161 | 0.060 | 0.161 | 0.060 | 0.161 | 0.000 | 0.000 | 0.061 | 0.163 | 0.061 | 0.162 | 0.061 | 0.161 | 0.061 | 0.162 | 0.000 | 0.001 |
| | 24 | 0.077 | 0.182 | 0.076 | 0.181 | 0.077 | 0.182 | 0.076 | 0.182 | 0.001 | 0.001 | 0.077 | 0.184 | 0.076 | 0.182 | 0.077 | 0.183 | 0.077 | 0.183 | 0.000 | 0.001 |
| | 48 | 0.104 | 0.215 | 0.107 | 0.215 | 0.110 | 0.216 | 0.107 | 0.215 | 0.003 | 0.001 | 0.110 | 0.218 | 0.107 | 0.215 | 0.109 | 0.215 | 0.109 | 0.216 | 0.001 | 0.002 |
| | 96 | 0.148 | 0.253 | 0.150 | 0.255 | 0.144 | 0.251 | 0.148 | 0.253 | 0.003 | 0.002 | 0.146 | 0.256 | 0.152 | 0.256 | 0.152 | 0.256 | 0.150 | 0.256 | 0.003 | 0.000 |
| PEMS04 | 12 | 0.067 | 0.166 | 0.067 | 0.167 | 0.066 | 0.166 | 0.067 | 0.167 | 0.000 | 0.000 | 0.069 | 0.170 | 0.067 | 0.167 | 0.067 | 0.166 | 0.068 | 0.168 | 0.001 | 0.002 |
| | 24 | 0.077 | 0.181 | 0.077 | 0.182 | 0.077 | 0.181 | 0.077 | 0.181 | 0.000 | 0.000 | 0.082 | 0.187 | 0.078 | 0.183 | 0.078 | 0.182 | 0.079 | 0.184 | 0.002 | 0.003 |
| | 48 | 0.097 | 0.206 | 0.095 | 0.205 | 0.096 | 0.205 | 0.096 | 0.206 | 0.001 | 0.001 | 0.103 | 0.213 | 0.097 | 0.207 | 0.096 | 0.206 | 0.099 | 0.209 | 0.003 | 0.004 |
| | 96 | 0.123 | 0.233 | 0.122 | 0.233 | 0.123 | 0.232 | 0.123 | 0.233 | 0.000 | 0.000 | 0.129 | 0.242 | 0.122 | 0.233 | 0.126 | 0.235 | 0.125 | 0.236 | 0.004 | 0.005 |
| PEMS07 | 12 | 0.051 | 0.143 | 0.052 | 0.143 | 0.052 | 0.144 | 0.052 | 0.144 | 0.000 | 0.000 | 0.053 | 0.147 | 0.052 | 0.144 | 0.051 | 0.143 | 0.052 | 0.145 | 0.001 | 0.002 |
| | 24 | 0.063 | 0.159 | 0.063 | 0.159 | 0.063 | 0.159 | 0.063 | 0.159 | 0.000 | 0.000 | 0.065 | 0.164 | 0.063 | 0.160 | 0.063 | 0.160 | 0.064 | 0.161 | 0.001 | 0.002 |
| | 48 | 0.081 | 0.179 | 0.080 | 0.180 | 0.081 | 0.180 | 0.080 | 0.180 | 0.000 | 0.000 | 0.084 | 0.188 | 0.081 | 0.181 | 0.080 | 0.180 | 0.082 | 0.183 | 0.002 | 0.004 |
| | 96 | 0.103 | 0.203 | 0.103 | 0.203 | 0.108 | 0.207 | 0.105 | 0.204 | 0.003 | 0.002 | 0.113 | 0.217 | 0.106 | 0.207 | 0.102 | 0.202 | 0.107 | 0.209 | 0.005 | 0.008 |
| PEMS08 | 12 | 0.071 | 0.170 | 0.072 | 0.170 | 0.070 | 0.170 | 0.071 | 0.170 | 0.001 | 0.000 | 0.073 | 0.174 | 0.071 | 0.170 | 0.070 | 0.169 | 0.071 | 0.171 | 0.002 | 0.003 |
| | 24 | 0.096 | 0.196 | 0.097 | 0.196 | 0.094 | 0.195 | 0.096 | 0.196 | 0.001 | 0.001 | 0.102 | 0.204 | 0.097 | 0.198 | 0.095 | 0.194 | 0.098 | 0.199 | 0.003 | 0.005 |
| | 48 | 0.149 | 0.244 | 0.152 | 0.247 | 0.150 | 0.245 | 0.150 | 0.245 | 0.001 | 0.002 | 0.166 | 0.259 | 0.155 | 0.249 | 0.155 | 0.249 | 0.158 | 0.252 | 0.006 | 0.006 |
| | 96 | 0.253 | 0.309 | 0.265 | 0.310 | 0.256 | 0.306 | 0.258 | 0.308 | 0.006 | 0.002 | 0.297 | 0.331 | 0.266 | 0.310 | 0.259 | 0.306 | 0.274 | 0.316 | 0.020 | 0.013 |
| Solar | 96 | 0.173 | 0.233 | 0.191 | 0.254 | 0.181 | 0.245 | 0.182 | 0.244 | 0.009 | 0.010 | 0.192 | 0.256 | 0.184 | 0.236 | 0.184 | 0.235 | 0.187 | 0.243 | 0.005 | 0.012 |
| | 192 | 0.199 | 0.257 | 0.191 | 0.250 | 0.191 | 0.254 | 0.194 | 0.254 | 0.004 | 0.003 | 0.196 | 0.248 | 0.191 | 0.252 | 0.193 | 0.255 | 0.193 | 0.252 | 0.002 | 0.003 |
| | 336 | 0.211 | 0.263 | 0.202 | 0.263 | 0.213 | 0.272 | 0.209 | 0.266 | 0.006 | 0.005 | 0.203 | 0.256 | 0.206 | 0.259 | 0.205 | 0.264 | 0.205 | 0.260 | 0.002 | 0.004 |
| | 720 | 0.209 | 0.270 | 0.217 | 0.270 | 0.220 | 0.269 | 0.216 | 0.270 | 0.006 | 0.001 | 0.208 | 0.260 | 0.207 | 0.265 | 0.214 | 0.271 | 0.210 | 0.265 | 0.004 | 0.005 |
| Traffic | 96 | 0.413 | 0.261 | 0.412 | 0.261 | 0.410 | 0.260 | 0.412 | 0.260 | 0.001 | 0.000 | 0.417 | 0.266 | 0.412 | 0.262 | 0.414 | 0.261 | 0.414 | 0.263 | 0.003 | 0.003 |
| | 192 | 0.432 | 0.271 | 0.433 | 0.271 | 0.431 | 0.270 | 0.432 | 0.271 | 0.001 | 0.001 | 0.434 | 0.276 | 0.432 | 0.272 | 0.431 | 0.270 | 0.432 | 0.273 | 0.002 | 0.003 |
| | 336 | 0.450 | 0.277 | 0.451 | 0.277 | 0.450 | 0.277 | 0.450 | 0.277 | 0.001 | 0.000 | 0.448 | 0.283 | 0.443 | 0.279 | 0.448 | 0.277 | 0.446 | 0.280 | 0.003 | 0.003 |
| | 720 | 0.486 | 0.295 | 0.493 | 0.297 | 0.485 | 0.296 | 0.488 | 0.296 | 0.004 | 0.001 | 0.491 | 0.301 | 0.481 | 0.296 | 0.477 | 0.296 | 0.483 | 0.298 | 0.007 | 0.003 |
| Weather | 96 | 0.157 | 0.200 | 0.158 | 0.201 | 0.158 | 0.201 | 0.158 | 0.201 | 0.001 | 0.001 | 0.157 | 0.200 | 0.160 | 0.204 | 0.158 | 0.201 | 0.158 | 0.202 | 0.002 | 0.002 |
| | 192 | 0.206 | 0.245 | 0.206 | 0.245 | 0.206 | 0.246 | 0.206 | 0.245 | 0.000 | 0.000 | 0.206 | 0.245 | 0.205 | 0.244 | 0.206 | 0.246 | 0.206 | 0.245 | 0.001 | 0.001 |
| | 336 | 0.262 | 0.287 | 0.262 | 0.287 | 0.264 | 0.288 | 0.263 | 0.288 | 0.001 | 0.001 | 0.262 | 0.287 | 0.264 | 0.288 | 0.264 | 0.289 | 0.263 | 0.288 | 0.001 | 0.001 |
| | 720 | 0.344 | 0.342 | 0.343 | 0.342 | 0.343 | 0.342 | 0.343 | 0.342 | 0.001 | 0.000 | 0.344 | 0.342 | 0.344 | 0.342 | 0.344 | 0.342 | 0.344 | 0.342 | 0.000 | 0.000 |

