# OpenReview forum: "Temporal Query Network for Efficient Multivariate Time Series Forecasting"
_ICML.cc/2025/Conference — ICML 2025 poster_

### Official Review · Reviewer_yR9p · 2025-03-07

**Overall Recommendation:** 3

**Summary:**

The paper presents a new model to do point-forecasting of multivariate time series: TQNet.

TQNet combines a single attention layer to handle multi-time-steps interactions and an MLP to handle multi-channel interactions. The main novel contribution of TQNet is that the attention layer doesn't use the input to create its query vectors, but instead use trained parameters based on the absolute time position modulo the data periodicity. Another particularity is that all channels in a single timestep is taken as a single "token", instead of having each timestep/channel pair identifying its own token.

The model is benchmarked against multiple recently published models on multiple multivariate datasets, and comes ahead.

**Claims And Evidence:**

The major portion of the evidence is done through the benchmark and the ablation experiments. The main evidence which is lacking is how good the model is at using multivariate information. While Figure 5 shows that TQNet is still better than baselines when the data is disrupted, this doesn't give much evidence that TQNet actively use multivariate information a lot. The results could just as well be explained by the baselines mostly having learned univariate information. Furthermore, this could also be explained by TQNet being relatively weak at using intra-channel information, and compensating this using inter-channel information.

**Essential References Not Discussed:**

The related work section only considers very recent models. It is therefore lacking in reference to classical statistical techniques, some of which having features that would deserve to be compared with TQNet. The main one would be the ETS technique, which also takes a strong advantage of being given the periodicity of the signal to improve a forecast.

**Experimental Designs Or Analyses:**

Nothing to mention besides what is already in "Methods And Evaluation Criteria".

**Methods And Evaluation Criteria:**

The benchmark contains many baselines and datasets, which gives credence to the point that TQNet is a useful model in practice.

However, there are three concerns about these results:
1. Many of the datasets used are known to be easily forecasted using univariate forecasting (such as the ETTh and Solar datasets). This makes it plausible that the results quality is solely due to the improved time-wise interactions.
2. Giving the model the periodicity of the dataset can be a very useful information which the model doesn't have to handle itself. It is not mentioned whether any of the baselines have also been given this information.
3. Classical and older models are absent from the baselines, with the oldest baseline being from 2022. At the very least, methods which strongly leverage the periodicity of the data should be included, such as ETS.

**Other Comments Or Suggestions:**

Please add the hyper parameters used for both TQNet and the baselines in the appendix. Additionally, if any hyper parameter search was done, it should be detailed in said appendix.

**Other Strengths And Weaknesses:**

Beside what is already mentioned in other sections, my main concern is that the quality of the results are solely due to TQNet strongly imposing a specific period to the forecast. To truly determine whether or not this is the case, further experiments should be added. I would personally suggest adding experiments with synthetic data, since they can be tailored to test specific property of the model.

**Questions For Authors:**

1. Is there an updated link to the anonymous code repository?
2. What is the impact of giving an incorrect W to the model? In particular, how does the model react when W is a multiple or a divisor of the full periodicity? An example would be to use 1 day instead of 1 week for the traffic dataset, or using 2 days instead of 1 day for the solar dataset.
3. If the query vectors are computed only through the time steps up to the period, does this mean that the interactions between time steps t and t' are identical to the interactions between time steps t and t'+W? If this is indeed the case, did you test how TQNet fare on datasets with a strong short-term causal interaction between time steps? One simple example could be a sine wave summed with a random walk.

**Relation To Broader Scientific Literature:**

The ablation where the paper shows the TQ attention can be applied to other architecture is a good sign that the ideas presented in the paper may be used to build new models. In particular, while I have doubts about it due to the lack of discrimination between interactions between time steps dt or dt+W time steps apart, the TQ attention could be mixed with other ways to encode temporal distance in a time series model.

**Theoretical Claims:**

There are no theoretical claim in the paper.

---

> ### Author Rebuttal · Authors · 2025-03-31
>
> **Thank you for your detailed and thoughtful review! We must apologize for the concise response earlier due to length limits.**
>
> > Summary: TQNet combines a single attention layer to handle multi-time-steps interactions and an MLP to handle multi-channel interactions.
>
> Sorry for any unclear points in our previous version. In fact, it is the **TQ-enhanced attention layer that handles multi-channel interactions**, followed by an MLP that handles multi-time-step interactions. We will improve the clarity of this statement in the revised paper.
>
> > Claims: The main evidence which is lacking is how good the model is at using multivariate information.
>
> Thank you for pointing this out. Figure 5 only indirectly demonstrates TQNet's ability to utilize multivariate information but does not directly show how effective it is in leveraging such information. To address this, we have added experiments on several large-scale multivariate datasets. In these experiments, the goal is to predict the target channel using multiple variables, ranging from not utilizing any additional variables to fully leveraging all available variables. **The results show that incorporating covariate information significantly improves the prediction accuracy of TQNet for the target channel.(even for Solar dataset)** This provides direct evidence that TQNet can effectively utilize multivariate information.
>
> | # covariant | Electricity | Traffic | Solar | PEMS03 | PEMS04 | PEMS07 |
> | - | - | - | - | - | - | - |
> | 0| 0.355| 0.153| 0.290|0.070|0.060| 0.125|
> | 5| 0.325| 0.150| 0.264|0.061|0.054| 0.088|
> | 20| 0.328| 0.149| 0.259|0.064|0.048| 0.081|
> | 100| 0.330| 0.138| 0.263|0.057|0.049| 0.075|
> | Full| 0.323| 0.137| 0.260|0.062|0.048| 0.079|
>
> > M1. This makes it plausible that the results' quality is solely due to the improved time-wise interactions.
>
> Our goal is to explore a more elegant mechanism for modeling multivariate correlations, for which we designed TQ-MHA. Time-wise interactions are not the primary focus of our study, so we adopted a simple two-layer MLP for temporal dependency modeling, which has been shown in recent studies to be sufficiently capable of capturing temporal dependencies. **As shown in Figure 2, TQNet's major improvements are particularly evident in large-scale multivariate datasets.**
>
> > M2. It is not mentioned whether any of the baselines have also been given this information.
>
> **Many baselines have indeed utilized temporal (time) information**. For example, CycleNet explicitly models periodic sequences, while iTransformer and TimeXer use timestamp sequences as tokens. *In fact, how to leverage this information to build better forecasting models remains an open research question, and TQNet provides an elegant solution in this regard.*
>
> > M3 (also mentioned in Essential References Not Discussed). Classical and older models are absent from the baselines, such as ETS.
>
> We have added comparison experiments with ETS, which clearly demonstrate the superiority of TQNet. Please refer to this link for the results: https://anonymous.4open.science/r/TQNet-ETS-CD4F.
>
> > Other Comments: Please add the hyperparameters used for TQNet.
>
> Yes, we have detailed the hyperparameters of TQNet in Appendix A.2, where they are set relatively consistently accross datasets. The results for other baselines are sourced from their official results or the iTransformer paper to ensure reliability.
>
> > Q1 (also mentioned in Supplementary Material):  The provided link does not work.
>
> We sincerely apologize for this issue. The anonymous website encountered a bug displaying "The requested file is not found" when trying to browse the code online. To resolve this, **you may directly download the source code from the anonymous website or access it through the Supplementary Material on OpenReview.** We hope this works for you.
>
> > Q2: What is the impact of giving an incorrect W to the model?
>
> In Figure 6, we have systematically explored the impact of incorrect W settings, except for the specific case you mentioned where W is set to a multiple of the full periodicity. **Our additional experiments show that when W = 2 × 168, TQNet's performance declines but remains close to that when W = 168.** This is primarily because setting W to an integer multiple only reduces the number of training samples allocated to TQ parameters proportionally, without significantly affecting the effectiveness of the TQ mechanism itself. We will add this result to Figure 6.
>
> || 168| 2*168 |
> | -| - | - |
> |MSE|0.164 | 0.167 |
> |MAE| 0.259 | 0.261 |
>
> > Q3: Whether interactions between time steps t and t' and those between t and t' + W are identical.
>
> They are not identical. **This is because the TQ mechanism only affects the correlation of Q in the attention mechanism, while K and V remain dependent on local samples.** Therefore, the fundamental purpose of the TQ mechanism is to provide a globally stable correlation supplement.
>
> **Apologies again for the concise text, and thank you again!**

---

> > ### Comment · Reviewer_yR9p · 2025-04-08
> >
> > I thank the authors for taking their time for the rebuttal. While I believe that the suggested clarifications and additions will improve their submission, it does not do it enough for me to increase my score.

---

### Official Review · Reviewer_thzz · 2025-03-11

**Overall Recommendation:** 4

**Summary:**

This paper introduces the Temporal Query Network (TQNet) to address multivariate time series forecasting (MTSF) tasks. At its core, it employs periodically shifted learnable parameters to model more stable inter-variable correlations adaptively. Extensive experiments are conducted to demonstrate the effectiveness of the proposed method.

**Claims And Evidence:**

Yes. The authors design lots of experimental evidence to support their claims.

**Essential References Not Discussed:**

Not at all.

**Experimental Designs Or Analyses:**

Yes. I have examined the experimental design.

**Methods And Evaluation Criteria:**

Yes. The proposed method is aligned to advance the field of MTSF.

**Other Comments Or Suggestions:**

1. There is an inconsistency in Algorithm 1 regarding the RevIN formula, which does not match Equation 8.
2. Figure 2 is missing the linear transformation mentioned in line 9 of Algorithm 1.

**Other Strengths And Weaknesses:**

**Strengths:**
1. The paper is well-organized and easy to follow.
2. The proposed method is logically sound and computationally efficient.
3. The experimental design is well-structured to demonstrate the effectiveness of the method, including ablation studies, exploratory experiments, and efficiency analysis.
4. The figures and tables are clear and visually appealing.


**Weaknesses:**
1. The paper only provides results for multivariate-to-multivariate forecasting. However, in real-world applications, a more common scenario is multivariate-to-univariate forecasting, where exogenous variables are used to predict a single target variable.
2. There are some inconsistencies between Figure 2, Algorithm 1, and their descriptions in the main text.
3. The paper does not discuss potential limitations, such as cases where there is no periodicity or no significant multivariate dependencies.

**Questions For Authors:**

1. The proposed technique heavily depends on the hyperparameter W. What happens when a suitable W cannot be found (i.e., when the dataset lacks clear periodicity)?
2. Why are the learnable parameters in TQ initialized to zero? How does this differ from random initialization?
3. Why can TQNet achieve nearly the same computational efficiency as DLinear (as shown in Figure 7)? This seems difficult to achieve, as the additional attention mechanism and deep network should require a significant amount of computation.
4. Are the results reported in Table 2 averaged over multiple runs? Were the baseline results reproduced, or were they sourced from existing work?

**Relation To Broader Scientific Literature:**

This paper proposes a multivariate time series forecasting model, which is particularly relevant to real-world applications such as traffic prediction, power demand forecasting, and weather forecasting.

**Theoretical Claims:**

No theoretical claims are made in the paper.

---

> ### Author Rebuttal · Authors · 2025-03-31
>
> **Thank you very much for your detailed review!**
>
> > W1: The paper only provides results for multivariate-to-multivariate forecasting.
>
> We have further supplemented the comparison between TQNet and baseline models in the multivariate-to-univariate forecasting scenario. **The results show that TQNet still exhibits a significant advantage in this setting**, demonstrating its superior capability in multivariate modeling.
>
> || TQNet|| TimeXer || iTrans || TimesNet || PatchTST ||
> | ----------- | --------- | --------- | ------- | ----- | ------ | ----- | -------- | ----- | -------- | ----- |
> || MSE| MAE| MSE| MAE| MSE| MAE| MSE| MAE| MSE| MAE|
> | ETTm2| **0.118** | **0.254** | 0.120   | 0.258 | 0.127  | 0.267 | 0.129    | 0.270 | 0.120    | 0.258 |
> | Electricity | **0.323** | **0.404** | 0.327   | 0.408 | 0.365  | 0.442 | 0.410    | 0.476 | 0.394    | 0.446 |
> | Traffic| **0.135** | **0.210** | 0.156   | 0.235 | 0.161  | 0.246 | 0.171    | 0.264 | 0.173    | 0.253 |
>
> > W2 (also mentioned in Suggestions): There are some inconsistencies between Figure 2, Algorithm 1, and their descriptions in the main text.
>
> We appreciate your careful examination and suggestions! We will correct these inconsistencies in the final version.
>
> > W3: The paper does not discuss potential limitations, such as cases where there is no periodicity or no significant multivariate dependencies.
>
> Thank you for highlighting this. **We will add a discussion of this potential limitations** of TQNet in the revised version to provide readers with a clearer understanding of TQNet and identify areas that require further investigation. Specifically, in scenarios where there is no significant periodicity, TQNet may not perform optimally, as its TQ mechanism relies on periodic shift operations. However, long-horizon forecasting in non-periodic settings is inherently challenging. In such cases, incorporating additional prior features may help compensate for the lack of periodic information. We will include these clarifications in the revised paper.
>
> > Q1: What happens when a suitable $W$ cannot be found?
>
> Many recent studies have demonstrated that **periodicity is a crucial factor for achieving long-horizon forecasting** [1]. Moreover, real-world time series data are often influenced by social or natural factors, meaning they typically exhibit at least a daily periodic fluctuation. Therefore, identifying a suitable $W$ is feasible and straightforward. Additionally, the results in Figure 6 indicate that even without leveraging the dataset's periodicity (e.g., setting $W$ to 1), **the TQ mechanism can still improve forecasting accuracy**. This is because, in such cases, TQ can act as an implicit channel identifier [2], enhancing the model's ability to distinguish between multivariate channels.
>
> [1] Lin S, Lin W, Hu X, et al. Cyclenet: Enhancing time series forecasting through modeling periodic patterns. NeurIPS, 2024.
> [2] Shao Z, Zhang Z, Wang F, et al. Spatial-temporal identity: A simple yet effective baseline for multivariate time series forecasting. CIKM, 2022.
>
> > Q2: Why are the learnable parameters in TQ initialized to zero?
>
> TQ can adaptively learn optimal representations of inter-variable relationships through backpropagation. Therefore, **its initialization does not significantly impact the final learned representations**. Specifically, we verified this by experimenting with different initialization strategies on the Electricity dataset and found that the performance remained consistent across different initialization methods.
>
> |      | Zero  | Uniform | Normal | Xavier | Kaiming |
> | ---- | ----- | ------- | ------ | ------ | ------- |
> | MSE  | 0.164 | 0.166   | 0.166  | 0.166  | 0.166   |
> | MAE  | 0.259 | 0.260   | 0.260  | 0.260  | 0.260   |
>
>
> > Q3: Why can TQNet achieve nearly the same computational efficiency as DLinear (as shown in Figure 7)?
>
> This is primarily due to two factors: (i) **The lightweight design of TQNe**t, which includes only a single attention layer and a two-layer MLP, maximizing efficiency. (ii) **The powerful parallel computing capabilities of modern GPUs**. Specifically, the attention computations and MLP structures in TQNet can be highly parallelized, allowing it to achieve computational efficiency comparable to linear models.
>
> > Q4: Are the results reported in Table 2 averaged over multiple runs?
>
> **Table 5 presents results from multiple runs of TQNet with different random seeds and learning rates, demonstrating its robustness**. The results in Table 2 for TQNet are from a single run with a random seed of 2024, following the experimental setup described in Appendix A.2. Notably, these results are consistent with those in Table 5. Furthermore, the baseline results in Table 2 are sourced from their official reports (or reproduced from the iTransformer paper) to ensure reliability. We have clarified this in the caption of Table 4, where the full results are provided.
>
> **Thanks again!**

---

> > ### Comment · Reviewer_thzz · 2025-04-03
> >
> > Thank you for your detailed response, which addressed my previous concerns. I have a further question regarding the rationale and objectives behind your method's lightweight design. Why does it utilize only a single-layer attention mechanism and a single MLP module? Would additional stacking (increasing capacity) lead to further improvements?

---

> > > ### Author Response · Authors · 2025-04-05
> > >
> > > **Dear Reviewer thzz,**
> > >
> > > Thank you very much for your feedback. We are delighted to hear that our previous rebuttal addressed your earlier concerns.
> > >
> > > > *I have a further question regarding the rationale and objectives behind your method's lightweight design. Why does it utilize only a single-layer attention mechanism and a single MLP module? Would additional stacking (increasing capacity) lead to further improvements?*
> > >
> > > Thank you for this insightful question. Indeed, the lightweight design is a key strength of our approach. TQNet achieves state-of-the-art performance using only the most essential components—namely, a single-layer attention mechanism and a single MLP—striking an optimal balance between forecasting accuracy and computational efficiency. This effectiveness is supported by two core factors:
> > >
> > > 1. **TQNet already incorporates all the essential components required for accurate time series forecasting**:
> > >     (i) the *Temporal Query* technique leverages periodic structures in time series;
> > >     (ii) the *TQ-enhanced attention mechanism* captures inter-variable (multivariate) dependencies;
> > >     (iii) the *MLP* effectively models temporal dependencies;
> > >     (iv) *RevIN* addresses distribution shifts commonly observed in time series data.
> > > 2. **We believe that time series forecasting should not be unnecessarily over-complicated**, and a well-designed, simple neural network is often sufficient for achieving strong performance [1][2][3].
> > >
> > > To further address your question, we conducted additional experiments by increasing the model capacity—specifically, stacking three layers of TQ-MHA and MLP modules.
> > >
> > > | Dataset     | Original-MSE | Stacking-MSE | Original-MAE | Stacking-MAE |
> > > | ----------- | ------------ | ------------ | ------------ | ------------ |
> > > | ETTh1       | **0.441**    | 0.443        | **0.434**    | 0.447        |
> > > | ETTh2       | **0.378**    | 0.382        | **0.402**    | 0.405        |
> > > | ETTm1       | **0.377**    | 0.391        | **0.393**    | 0.404        |
> > > | ETTm2       | **0.277**    | 0.280        | **0.323**    | 0.324        |
> > > | Electricity | **0.164**    | 0.167        | **0.259**    | 0.262        |
> > > | Solar       | **0.198**    | 0.203        | **0.256**    | 0.265        |
> > > | traffic     | **0.445**    | 0.451        | **0.276**    | 0.285        |
> > > | weather     | **0.242**    | 0.242        | **0.269**    | 0.271        |
> > > | PEMS03      | 0.097        | **0.095**    | 0.203        | **0.197**    |
> > > | PEMS04      | 0.091        | **0.084**    | 0.197        | **0.189**    |
> > > | PEMS07      | 0.075        | **0.072**    | 0.171        | **0.167**    |
> > > | PEMS08      | **0.142**    | 0.147        | 0.229        | **0.225**    |
> > >
> > > The results show that increasing the depth does **not lead to significant improvements**. In fact, for most datasets, performance slightly decreases, while some improvement is observed on the PEMS datasets. This outcome highlights the **robustness and rationality of the original TQNet design**, which already achieves near-optimal performance with minimal architectural complexity.
> > >
> > > This not only validates the effectiveness of our method but also supports our central claim: **TQNet provides an ideal trade-off between forecasting performance and computational cost**. We advocate for lightweight designs in time series forecasting, as they facilitate interpretability, enable easier deployment in practical applications, and remain competitive in accuracy—all of which constitute major advantages of TQNet.
> > >
> > > Once again, thank you for your thoughtful question. We hope this response addresses your further concerns.
> > >
> > >
> > >
> > > [1] Lin S, Lin W, Wu W, et al. SparseTSF: Modeling Long-term Time Series Forecasting with *1k* Parameters. Forty-first International Conference on Machine Learning (ICML), 2024.
> > >
> > > [2] Xu Z, Zeng A, Xu Q. FITS: Modeling Time Series with $10 k$ Parameters. The Twelfth International Conference on Learning Representations (ICLR), 2024.
> > >
> > > [3] Zeng A, Chen M, Zhang L, et al. Are transformers effective for time series forecasting? Proceedings of the AAAI conference on artificial intelligence (AAAI) 2023.

---

### Official Review · Reviewer_Tew6 · 2025-03-12

**Overall Recommendation:** 2

**Summary:**

This paper proposed Temporal Query technique for multivariate time series forecasting framework, which is aiming at capturing optimal representations of inter-variable relationships. The lightweight improvement show advanced performance on real-world datasets and can be integrated easily to existing models.

**Claims And Evidence:**

Evidence insufficient. In Figure 1, the author's viewpoint is that the inter-variable correlations observed in individual samples is significantly differ from global correlations because of non-stationary disturbances, such as extreme values, missing data, and noise. However, more intuitive reason may be that the correlation varies across different time scales[1] even if there are no extreme values, missing data, and noise.

[1] MSGNet: Learning Multi-Scale Inter-Series Correlations for Multivariate Time Series Forecasting.

**Essential References Not Discussed:**

The following references may help for the similar motivation, which is to discuss the variate correlations:
[1] MSGNet: Learning Multi-Scale Inter-Series Correlations for Multivariate Time Series Forecasting.

**Experimental Designs Or Analyses:**

The experiments is conducted with commonly used settings and full results of repeated experiments are shown in appendix.

**Methods And Evaluation Criteria:**

1. Lightweight and exquisite improvement. The TQ technique consumes smaller parameter sizes and shorter training times.
2. Good portability. The TQ technique can be integrated into several time series forecasting models easily.
3. Sufficient evaluations.
(1) State-of-the-art performance for long-term forecasting. The model achieves state-of-the-art performance on some real-world multivariate datasets.
(2) Experiments on representation learning suggests that the TQ technique is useful for captureing intrinsic correlations among different channels.
(3) The dependency study is interesting, which evaluate whether the TQ technique captures more robust multivariate dependencies.

**Other Comments Or Suggestions:**

N/A

**Other Strengths And Weaknesses:**

N/A

**Questions For Authors:**

1.How does temporal query technique benefit representation learning of inter-variable relationships? Why can TQ-enhanced MHA mechanism learns a globally consistent and adaptive representation of inter-variable correlations within individual samples? How does it deal with non-stationary disturbances in real-world data? Maybe some theoretical explanation or case discussion will help.

**Relation To Broader Scientific Literature:**

This work contributes to improving the predictive performance of time series models and will be highly effective in low-cost industrial scenarios.

**Theoretical Claims:**

Lack or insufficient theoretical analysis. The article lacks theoretical discussion on how can TQ address the non-stationary disturbances in real-world data and how it finally enhances the robustness of the learned correlations. This doesn't sound intuitive, and there’s no theoretical explanation or case discussion.

---

> ### Author Rebuttal · Authors · 2025-03-31
>
> **Thank you for your insightful review!**
>
> > Claims And Evidence and Essential References Not Discussed.
>
> Thanks for pointing this out. Indeed, changes in time scales can cause variations in inter-variable correlations, and in fact, Figure 1 of TQNet also demonstrates this. The key differences are:
>
> 1. MSGNet's multi-scale approach focuses on **different time scales within the look-back window** (e.g., variations within 24 and 96-time steps in a window of length 96).
> 2. TQNet's multi-scale nature manifests in two ways: **the local sample look-back window** represents short-term scales, while **the global sequence (the entire training set)** represents long-term scales.
>
> **Thus, MSGNet’s findings actually support our claim that considering correlations at different scales is necessary.** *We will supplement the revised paper with discussions and comparisons with MSGNet.* The complete comparison results can be found at this link: https://anonymous.4open.science/r/TQNet-MSGNet-7EBE, which demonstrates the significant advantage of TQNet.
>
> > Questions For Authors: How does temporal query technique benefit representation learning of inter-variable relationships? Why can TQ-enhanced MHA mechanism learns a globally consistent and adaptive representation of inter-variable correlations within individual samples? How does it deal with non-stationary disturbances in real-world data? Maybe some theoretical explanation or case discussion will help.
>
> The benefits of the Temporal Query (TQ) technique arise from **its learnable nature combined with a periodic shifting mechanism**. In the TQ-enhanced MHA, the query $Q$ is generated from globally shared, periodically shifted learnable vectors, while the keys $K$ and values $V$ are directly derived from the raw time series and thus capture more localized correlations. During training, the model is encouraged to maximize the attention output:
>
> $$
> O = \operatorname{Softmax}\left( \frac{Q_i K_i^\top}{\sqrt{L}} \right) V_i,
> $$
>
> which implies that for each sample $i$, *the learned query $Q_i$ tends to align with the key $K_i$, meaning that it actively incorporates more relevant inter-variable information for accurate forecasting*. This alignment can be formulated as:
>
> $$
> \operatorname{Corr}(Q_i) \approx \operatorname{Corr}(K_i) = \frac{Q_i K_i^\top}{\|Q_i\|\,\|K_i\|},
> $$
>
> where $\operatorname{Corr}(\cdot)$ represents a normalized measure of correlation between the vectors.
>
> Moreover, since the query $Q_i$ is periodically extracted from the shared learnable parameter $\theta_{\text{TQ}}$ (i.e., multiple samples share the same $Q$ due to the periodic shift), its effective correlation remains consistent across periods and is averaged over multiple local samples.
>
> $$
> \operatorname{Corr}(Q_i) = \operatorname{Corr}(Q_{i+nW}), \quad n=0,1,\dots,N-1,
> $$
>
> and
>
> $$
> \operatorname{Corr}(Q_i) \approx \frac{1}{N} \sum_{n=0}^{N-1} \operatorname{Corr}(K_{i+nW}),
> $$
>
> where $W$ is the periodic length, $N$ is the number of sampled periods, and $K_{i+nW}$ represents the keys obtained from the raw time series, capturing localized correlations.
>
> **Thus, in practice, $Q_i$ serves as an averaged representation of correlations across all samples within the dataset over multiple periods**, *mitigating the impact of non-stationary disturbances in local samples.)*
>
> *This also explains why the TQ technique enables each sample's $Q$​ to learn a globally consistent and adaptive representation of inter-variable correlations, addressing the limitation of conventional attention mechanisms, which can only capture localized sample correlations. Since the learned correlations incorporate information from all periodic samples in the training set, they approximate the overall dataset correlation, thereby enhancing the representation learning of inter-variable relationships.*
>
> > Theoretical Claims.
>
> We hope the above theoretical analysis addresses your concerns. In summary, the TQ technique, **through its periodic cyclic sharing mechanism, effectively neutralizes non-stationary disturbances across multiple periods**, thereby improving the robustness of the correlation learned in the attention mechanism.
>
> **Additionally, Figure 1 serves as a case study demonstrating the effectiveness of the TQ technique.** The data is collected from the first sample of the real-world Traffic dataset (where Figure 1d highlights several channels with strong disturbances). It can be observed that even in the presence of noisy disturbances, the correlation of $Q$ generated by the TQ technique is more stable and aligns more closely with the dataset's global correlation. This characteristic enables TQ-MHA to comprehensively consider correlations at different time scales—where $Q$ models the global correlation via cross-period contributions, while $K$ and $V$ model localized correlation with noisy perturbations.
>
> **Finally, thank you again for your informative review. We hope our response addresses your concerns.**

---

### Official Review · Reviewer_kXGG · 2025-03-14

**Overall Recommendation:** 2

**Summary:**

This paper proposes Temporal Query Network (TQNet), a new approach for multivariate time series forecasting (MTSF). The key idea is the Temporal Query (TQ) technique, where periodically shifted learnable vectors serve as the query in a single-layer multi-head attention (MHA) module. TQ provides one vector per channel and shift them by a chosen cycle length. This mechanism aims to capture global inter-variable correlations more robustly than conventional self-attention, which often derives queries/keys/values solely from the raw data and is prone to get influenced by noise or missing values. The authors demonstrate state-of-the-art performance on 12 real-world benchmarking datasets of 3 different domains, showing that TQNet is both accurate and efficient, even with a large number of channels (e.g., nearly 1,000).

**Claims And Evidence:**

The main claims of this paper are (1). TQ can better capture inter-variable correlations, (2). TQ is more robust to noise and missing values, and (3). TQ is more efficient. While claim (2) and (3) seem to be intuitively reasonable, claim (1). is non-trivial and requires more convincing intuition and direct evidence.

The problems to be clarified include but not limited to:

C1. TQ is fixed across different samples. However, as shown in figure 1(a) and 1(c), the global correlations and the per-example correlations can be different, which can reveal some properties of each example. It is not clear how much this difference will play a role in MTSF, and how well TQ can tackle this problem

C2. As TQ generating vectors from CxW learnable space, it is not clear how the correlations among different variables can be learned - there seem to be no explicit constrains on the inter-variable correlations

C3. TQ seem to violate the invariance of the patterns in the time dimension, i.e., patten alpha in channel 1 and pattern beta in channel 2 might happen in different timestamps. However, in TQ, as each different timestamp will take a fixed CxL vector, patterns seem to be fixed in the time domain.

**Essential References Not Discussed:**

To my knowledge, this paper refers to a reasonably good number of references.

**Experimental Designs Or Analyses:**

The experiment designs, i.e., main results, ablation studies, MSE vs. Training efficient, sensitivity of W, against number of variables, seem reasonably complete and can empirically back up the advantage of the proposed method.

**Methods And Evaluation Criteria:**

Though the main claims of the method need to be further polished and supported, the methods follows the main claims well.

The datasets, the experimental settings (lock-back length and forecasting length) and the main evaluation metrics (MSE and MAE, training time, etc.) follow the common practice in MTSF.

**Other Comments Or Suggestions:**

C1. The message of Figure 1 is not clear. I suggest the authors to clearly put the message in the figure caption. For example, the message by comparing Figure 1(a) and 1(b) is that TQ can replicate well the global correlation.

**Other Strengths And Weaknesses:**

S1. The paper is largely well-written and easy to follow.

Besides the claim questions C1, C2 and C3 to be clarified, some other weaknesses include:

W1. The periodic length W is dataset-dependent, and there is no clearly way to easily and automatically determine W given a specific dataset.

W2. The fact that there is only one periodic length W for each dataset, might overlook the cases where there are multiple periodic lengths, e.g., hourly, daily, weekly and monthly, etc.

**Questions For Authors:**

Please refer to C1, C2, C3 and W1, W2.

**Relation To Broader Scientific Literature:**

To my understanding, there are no clear relations to a potential broader scientific literature, besides MTSF.

**Theoretical Claims:**

To my understanding, there are no theoretical claims in this paper.

---

> ### Author Rebuttal · Authors · 2025-03-31
>
> **Thank you for your valuable comments!**
>
> > C1: TQ is fixed across different samples, whereas per-example correlations differ from global correlations.
>
> This concern is valid in general, but it is effectively addressed within the TQ-MHA mechanism. In TQ-MHA, **the learnable shifted TQ serves *only* as the $Q$**, while the raw series acts as the $K$ and $V$. This design means that the correlations among **$Q$  emphasize more stable global dependencies**, while those among **$K$ and $V$ focus on per-example correlations**. Through the attention mechanism, the model integrates both global and local dependencies, rather than relying solely on per-example correlations. *We will properly revise the original claim to clarify this point.*
>
> > C2: How does TQ capture inter-variable correlations if there are no explicit constraints?
>
> In fact, TQ introduces **two implicit biases** that facilitate the learning of inter-variable correlations:
>
> 1. **Adaptive learning through backpropagation** – TQ is inherently designed to learn correlations via backpropagation optimization. As shown in Figure 4 of the paper, TQ effectively models the relative relationships among variables.
> 2. **Periodic shifting mechanism** – This ensures that TQ-based learned representations are averaged per period, mitigating the effects of random noise perturbations. Figure 1 demonstrates that the correlations derived from TQ-generated $Q$ are more stable and globally consistent.
>
> > C3: TQ seems to violate time-invariance because it assigns fixed ( C × L ) vectors to each timestamp.
>
> In fact, **only sample on timestamps $t$ and $t+W$ share the same $Q$**, since $Q$ ($C×L$) is extracted from $\theta_{TQ}$ ($C×W$) via a periodic shifting mechanism (with a period of $W$ timesteps). However, **the remaining samples within the interval $[t, t+W-1]$, the $Q$ are different** across timestamps (manifesting as a shift along the time axis). Therefore, this setting does not violate time-invariance. On the contrary, by explicitly considering periodic fluctuations, it better aligns with the inherent biases of real-world time series.
>
> > W1:  There is no clearly way to easily and automatically determine determine W given a specific dataset..
>
> There are many simple methods to clearly determine the hyperparameter $W$. On one hand, real-world time series data are often influenced by **social or natural factors**, meaning they typically exhibit **at least a daily or weekly periodic fluctuation**. Therefore, considering the sampling interval of the data (e.g., 15 min), it is easy to infer the potential periodic length. On the other hand, the **autocorrelation function (ACF) serves as an effective mathematical tool to verify the periodic length**, where peaks in the ACF correspond to potential periodic lengths. Therefore, adjusting and selecting $W$ is even easier than tuning the learning rate in the practical training scenarios. *We will provide a more detailed explanation of how to determine $W$ in the revised paper, along with a script tool (utilizing ACF) in the open-source code to facilitate direct usage by users.*
>
> > W2: It might overlook the cases where there are multiple periodic lengths.
>
> Thanks for pointing this out. This issue can be discussed in two cases:
>
> 1. **When multiple periodicities overlap, it can be easily handled**. For example, the Traffic dataset exhibits both daily (24-hour) and weekly (7×24-hour) periodic patterns. In this case, considering only the longest periodicity (weekly) is sufficient, as it inherently encompasses the shorter daily cycle. **In fact, most real-world scenarios fall into this category, so this does not pose a significant challenge for the practical application of TQNet**.
> 2. **When multiple periodicities are irregularly interwoven**, such as weekly (7×24-hour) and monthly (30×24-hour) cycles, it introduces some challenges for TQNet. Simply considering the longest periodicity cannot precisely capture the overlapping weekly patterns. In such cases, a practical compromise is to focus only on the daily periodicity (24-hour). Additionally, an alternative approach is to integrate multiple TQNet models, each configured with different $W$ values, to enhance adaptability in such scenarios. **Overall, fully addressing this issue remains an open research question, even for existing models.** *We will include a more detailed discussion in the revised paper to provide readers with a clearer understanding of TQNet and suggest directions for further investigation.*
>
> > Suggestion to clearly put the message in the Figure 1 caption.
>
> Thanks for your suggestion! *We will add a clearer message to the caption of Figure 1.*
>
> Finally, sorry about the concise response due to length limits. For any unclear points, we are happy to provide further clarification in the next stage. **Thanks again!**

---

> > ### Comment · Reviewer_kXGG · 2025-04-05
> >
> > Thanks for your detailed response. I think fixing W1 can be a very good improvement. The discussion of W2 also addresses my concern. However, I am not fully convinced by the responses to C1 to C2. After careful consideration, I will remain my score.

---

> > > ### Author Response · Authors · 2025-04-08
> > >
> > > **Dear Reviewer kXGG,**
> > >
> > > Thank you for your valuable feedback, and we sincerely apologize that our previous response did not fully address your concern. To address this properly, **we conducted new experiments and visualizations to better illustrate the working principles behind the TQ mechanism.**
> > >
> > > > C1. As shown in Figure 1(a) and 1(c), the global correlations and the per-example correlations can be different, which can reveal some properties of each example. **It is not clear how much this difference will play a role in MTSF, and how well TQ can tackle this problem.**
> > >
> > > Indeed, per-example correlations play an important role in MTSF. However, as in traditional approaches, considering only per-example correlations is insufficient. Therefore, we proposed TQ-MHA which utilizes a learnable vector as Query ($Q$) to model global correlations, while the Key ($K$) are derived from raw data to capture per-example correlations.
> > >
> > > To evaluate the difference between global and local correlations, and to demonstrate how well TQ can handle this issue, we conducted additional experiments. Specifically, we compared the following three scenarios:
> > >
> > > 1. **Both $Q$ and $K$ are generated from raw data**, capturing only per-example correlations. This is the traditional approach.
> > > 2. **$Q$ is generated from the learnable TQ vector, while $K$ is generated from raw data**. This allows the attention score computation $\text{Score} = \frac{QK^\top}{\sqrt{d}}$ to incorporate both global and per-example correlations. This is the method used in our current TQNet.
> > > 3. **Both $Q$ and $K$ are generated from the learnable TQ vector**, such that the attention score focuses solely on global correlations without considering local ones.
> > >
> > > The table below reports the average results across four forecast horizons on large-scale multivariate datasets. **As shown, considering both global and per-sample correlations (i.e., the TQNet strategy) yields the best performance, followed by using only per-sample or global correlations.**
> > >
> > > || (Q=Raw, |K=Raw)| (Q=TQ,|K=Raw)| (Q=TQ, |K=TQ)|
> > > | - | - | - | - | - | - | - |
> > > || MSE| MAE   | MSE| MAE   | MSE| MAE|
> > > | Electricity | 0.175| 0.267 | **0.164**| **0.259** | 0.179| 0.269 |
> > > | Solar| 0.208| 0.257 | **0.198**| **0.256** | 0.213| 0.268 |
> > > | Traffic| **0.426**| 0.279 | 0.445| **0.276** | 0.429| 0.281 |
> > > | PEMS03| 0.114| 0.222 | **0.097**| **0.203** | 0.111| 0.221 |
> > > | PEMS04| 0.112| 0.222 | **0.091**| **0.197** | 0.113| 0.222 |
> > > | PEMS07| 0.094| 0.195 | **0.075**| **0.171** | 0.092| 0.195 |
> > > | PEMS08| 0.170| 0.252 | **0.142**| **0.229** | 0.174| 0.257 |
> > >
> > > > C2. As TQ generating vectors from C×W learnable space, **it is not clear how the correlations among different variables can be learned**—there seem to be no explicit constraints on the inter-variable correlations.
> > >
> > > We apologize for our earlier misunderstanding. We now understand that your concern lies in how TQ learns inter-variable correlations without explicitly modeling variable structures (e.g., using graph structures).
> > >
> > > **In fact, this is handled by the attention mechanism.** During training, TQNet is optimized to maximize the attention output:
> > >
> > > $$ O = \operatorname{Softmax}\left( \frac{Q_i K_i^\top}{\sqrt{L}} \right) V_i, $$
> > >
> > > which means that for each sample $i$, the learned query $Q_i$ is encouraged to align with key $K_i$, thus incorporating more relevant inter-variable information for accurate forecasting. This alignment can be approximately formulated as:
> > >
> > > $$ \operatorname{Corr}(Q_i) \approx \operatorname{Corr}(K_i) = \frac{Q_i K_i^\top}{|Q_i||K_i|} $$
> > >
> > > Moreover, since the query $Q_i$ is periodically extracted from the shared learnable parameter $\theta_{\text{TQ}}$ (i.e., multiple samples share the same $Q$ due to periodic shift), the effective correlation remains consistent across different periods and is averaged over multiple local samples:
> > >
> > > $$ \operatorname{Corr}(Q_i) \approx \frac{1}{N} \sum_{n=0}^{N-1} \operatorname{Corr}(K_{i+nW}). $$
> > >
> > > Therefore, after sufficient training, the learned correlations in $Q$ implicitly incorporate information from all periodic samples in the dataset, effectively approximating the global dataset correlation. **In summary, it is the interaction enabled by the attention mechanism between $Q$ and $K$ that endows TQ with the ability to approximate global correlations.**
> > >
> > > To verify this, we performed a new experiment on the Traffic dataset by applying different Dropout rates to the attention scores **(Figure link: https://anonymous.4open.science/r/TQNet-Visual-4DB7)**. The results show that smaller Dropout rates (i.e., more interaction between $Q$ and $K$) lead to the learned TQ correlations that more closely resemble global correlations. **This further demonstrates that it is the attention mechanism’s interaction between Q and K that enables TQ to learn meaningful inter-variable representations.**
> > >
> > > **Thank you again for your thoughtful review. We hope our further explanation and evidence resolves your concerns.**

---

### Decision · Program_Chairs · 2025-05-01

**Decision:**

Accept (poster)

**Comment:**

This paper proposes the Temporal Query Network (TQNet) for multivariate time series forecasting. Its core innovation is the Temporal Query (TQ) mechanism, which uses periodically shifted learnable vectors as the query (Q) in a single-layer attention module, while the keys (K) and values (V) are derived from the input data. This design aims to capture stable, global inter-variable correlations more robustly and efficiently than traditional self-attention, especially in the presence of noise or non-stationarity. Reviewers acknowledged the model's strong performance on benchmarks, its efficiency, and the novelty of the TQ idea. However, several concerns were raised, particularly regarding the fundamental justification and intuition behind TQ: how fixed (though periodically shifted) queries capture dynamic per-sample correlations (R-kXGG), how inter-variable correlations are learned without explicit constraints (R-kXGG, R-Tew6), potential time-invariance issues (R-kXGG), and the lack of theoretical grounding for robustness claims (R-Tew6). Other key issues included the reliance on a manually set period (W), handling multiple periodicities (R-kXGG), insufficient direct evidence of effective multivariate information use (R-yR9p), the need for comparisons with classical methods like ETS (R-yR9p), and minor clarity issues in figures and algorithms (R-thzz, R-kXGG).

The authors provided extensive rebuttals, including theoretical arguments, clarifications, and significant new experimental results. They explained that TQ captures global patterns while K/V capture local ones, with the attention mechanism integrating both. They argued TQ learns correlations implicitly via optimization and averaging effects from periodic shifting, mitigating noise. They provided new experiments comparing different Q/K origins, showing TQ(Q)/Raw(K) was optimal, and experiments demonstrating how TQ learns global correlations through Q/K interaction. They addressed periodicity concerns by suggesting methods (ACF) to determine W, discussing multi-periodicity handling, and showing robustness to incorrect W values. New experiments demonstrated improved performance with more covariates, multivariate-to-univariate results, and comparisons against ETS and MSGNet. Reviewer responses were mixed: R-thzz was fully convinced, raising their score to Accept after follow-up questions on the model's lightweight design were answered. R-kXGG remained unconvinced by the core TQ explanations and maintained Weak Reject. R-Tew6 acknowledged the rebuttal but did not change their Weak Reject score or provide further feedback. R-yR9p acknowledged the improvements but felt they weren't substantial enough to raise their Weak Accept score.